

# Multivariate analysis of Kelvin wave seasonal variability in ECMWF L91 analyses

Marten Blaauw [1] and Nedjeljka Žagar [1]

[1]University of Ljubljana, Faculty of mathematics and physics, Ljubljana, Slovenia

*Correspondence to:* Marten Blaauw (marten.blaauw@fmf.uni-lj.si)

**Abstract.** The paper presents the seasonal variability of Kelvin waves (KWs) in 2007-2013 ECMWF analyses on 91 model
levels. The waves are filtered using the normal-mode function decomposition which simultaneously analyses wind and mass
field based on their relationships from linear wave theory. Both spectral as well as spatiotemporal features of the KWs are
examined in terms of their seasonal variability in comparison with background wind and stability. Furthermore, a differentiation
is made using spectral bandpass filtering between the slow horizontal barotropic KW response and the fast vertical projection
response observed as vertically-propagating KWs.
Results show a clear seasonal cycle in KW activity which is predominantly at the largest zonal scales (wavenumber 1-2)
where up to $50\%$ more energy is observed during the solstice seasons in comparison with spring and autumn. The spatiotempo-
ral structure of the KW reveals the slow response as a robust "Gill-type" structure with its position determined by the location
of the dominant convective outflow winds throughout the seasons. Its maximum strength occurs during northern summer when
easterlies in the Eastern Hemisphere are strongest. The fast response in the form of free traveling KWs occur throughout the
year with seasonal variability mostly found in the wave amplitudes being dependent on background easterly winds.



## 1 Introduction

Atmospheric equatorial Kelvin waves (hereafter KWs), first discovered in the stratosphere (Wallace and Kousky, 1968), are nowadays observed and studied over a broad range of spatial and temporal scales. A broad wavenumber-frequency spectrum can be traced to the spatiotemporal nature of tropical convection which generates KWs along with a spectrum of other equatorial waves. Atmospheric wave response to the stochastic nature of convection was studied by Garcia and Salby (1987) and Salby and Garcia (1987) who made a distinction between (i) projection or vertical response to short-term heating fluctuations (e.g. daily convection) and (ii) barotropic or horizontal response to seasonal convective heating. For KWs, the vertical response gives rise to a broad frequency spectrum of vertically propagating KWs that radiate outward into the stratosphere where they drive zonal-mean quasi-periodic flows such as the quasi-biennial oscillation (QBO, Holton and Lindzen, 1972). The horizontal response to seasonal transitions in convective heating gives rise to planetary-scale disturbances with a half-sinusoidal vertical structure confined to the troposphere. A part of this response remains stationary over the convective hotspot; its shape resembling a classic "Gill-type" KW solution (Gill, 1980). The other part of the response intensifies and advances over the Pacific, representing a transient component of the Walker circulation (Salby and Garcia, 1987).

Both components of the KW response received increased attention in the scientific community over the last decades in terms of the role they play in the (intra)seasonal variability of the Tropical Tropopause Layer (hereafter TTL), defined as a transition layer between the typical level of convective outflow at ∼12 km where the Brunt-Väisälä frequency is at its minimum, and the cold point tropopause at ∼16-17 km (Highwood and Hoskins, 1998; Fueglistaler et al., 2009). Within the TTL, temperature variations play an important role in controlling the stratosphere-troposphere exchange of various species such as ozone and water vapour thereby aiding in the dehydration process of air entering the stratosphere. The two parts of the KW response alternate the TTL differently on different time scales (Highwood and Hoskins, 1998; Randel and Wu, 2005; Ryu et al., 2008; Flannaghan and Fueglistaler, 2013); their relative contribution to TTL dynamics varies with season and is not yet fully understood. The present study contributes to this topic by applying a novel multivariate analysis of Kelvin wave seasonal variability in model-level analysis data.

Seasonal variations of Kelvin wave dynamics in the TTL have been previously studied using temperature data derived from satellites such as SABER (Sounding of the Atmosphere using Broadband Emission Radiometry, Garcia et al., 2005; Ern et al., 2008; Ern and Preusse, 2009), HIRDLS (High Resolution Dynamics Limb Sounder, Alexander and Ortland, 2010), and GPS-RO (Global Positioning System Radio Occultation, Tsai et al., 2004; Randel and Wu, 2005; Ratnam et al., 2006). For example, Alexander and Ortland (2010) reported a clear seasonal cycle around 16-17 km (∼ 100 hPa) in KW temperature observed by HIRDLS, coinciding closely with variations in background stability. A widely used method for the KW filtering from gridded data is the space-time spectral analysis introduced by Hayashi (1982). It operates on single variable data and it has been widely used to diagnose equatorial waves in the outgoing longwave radiation (OLR, e.g. Wheeler and Kiladis, 1999) and climate model outputs (e.g. Lin and Coauthors, 2006). Based on 40-year ECMWF reanalysis (ERA-40) data, Suzuki and Shiotani (2008) found that the temperature component of Kelvin waves tends to peak at 70 hPa while the zonal wind peaks at lower altitudes, i.e. at 100 hPa (150 hPa) in Eastern (Western) hemisphere.





The zonal wind and geopotential height of the KW are closely related. For a single zonal wavenumber $k$, the geopotential,
$\Phi_{kw}$, and the zonal wind $U_{kw}$ of a zonally propagating KW are related according to the following equation:
$$\Phi_{kw} = g\,h_{kw} = \frac{\nu}{k} U_{kw}, \qquad \text{where} \qquad U_{kw} = U_0 \exp(-\frac{\beta k y^2}{2\nu}). \qquad (1)$$
Here, $U_0$ is the KW amplitude in zonal wind on the equator, $\beta = 2\Omega/a$ ($\Omega$ being the rotation rate and $a$ the radius of Earth), $\nu$
is the wave frequency, $g$ is gravity and $y$ is the distance from the equator. These expressions are obtained as a special solution
of the linearized shallow-water equations on the equatorial $\beta-$plane (e.g. Holton, 2004, Chapter 11). The KW $e-$folding
decay width, $a_e$, is given by $a_e = (c/2\beta)^{1/2}$, where the KW phase speed $c$ is determined from the dispersion relation $\nu = kc$.
By prescribing the value of KW phase speed $c$ (i.e. the equivalent depth of the shallow-water equation system), analytical
solutions from linear wave theory can be used to simultaneously analyze wind and height data of the KW wave on a single
horizontal level. Such multivariate analysis was carried out by Tindall et al. (2006) who analyzed several levels the ECMWF
15-year reanalysis dataset (ERA-15) in the lower stratosphere. They reported a maximum of Kelvin wave activity at 100 hPa
around the solstices when tropical cloud activity maximizes. For the ERA-15 data in 1981-93 period, their Kelvin wave analysis
explained approximately 1 K$^2$ of temperature variance on the equator at 100 hPa.
The present paper extends the use of linear wave theory from the equatorial $\beta-$plane to the three-dimensional (3D) spherical
coordinates in order to analyze KW wind and temperature fields in recent ECMWF operational analyses. We focus on seasonal
variability of KWs in the TTL layer in the ECMWF operational analyses during a period when the model employed 91 vertical
level (L91) between the surface and 1 Pa. The L91 model was in operations between 2006 and early summer 2013 when it
was replaced by 137 levels. This study thus explores most of information on the vertical structure of KWs available in the
L91 analysis data. We present a methodology for the simultaneous analysis of wind and temperature perturbations associated
with KWs with respect to the background state and apply it to quantify scale-dependent seasonal KW variability in several
frequency bands.
The paper consists of five sections. Methodology of the KW diagnosis and the data are presented in section 2. Section 3
presents the KW energetics in wavenumber space focusing on the seasonal cycle. Section 4 presents a 3D view on KWs in
L91 dataset, both for the horizontal as well as for the vertical projection KW response. Conclusions and outlook are given in
section 5.
## 2   Data and methodology
The Kelvin waves are filtered using the Normal-Mode Function (NMF) decomposition derived by Kasahara and Puri (1981) and
briefly summarized below. Input ECMWF operational analyses cover 6 years from January 2007 till June 2013, approximately
6.5 years. The dataset starts after two important updates in the ECMWF assimilation cycle: a resolution update on 1 February
2006 and the introduction of GPS-RO temperature profiles in the assimilation on 12 December 2006. The data ends at the
next update in vertical resolution from L91 to L137 on 25 June 2013. A case study of the large-scale KW in July 2007 (Žagar





et al., 2009) showed how the NMF method provides information on the horizontal and vertical wave structure and its vertical
propagation in the stratosphere.

## 2.1   Filtering of Kelvin waves by 3D normal-mode function expansion

The basic assumption behind the NMF expansion is that a global state of the atmosphere described by its mass and wind vari-
ables at any time can be considered as a superposition of the linear wave solutions upon a predefined background state. These
linear solutions describe two types of wave motions: Rossby waves and inertio-gravity waves which obey their corresponding
dispersion relationships. The sssociated eigensolutions in terms of the Hough harmonics define both mass and wind fields of
the waves. The linear wave theory approach has been successfully employed in many studies, especially for the large-scale
tropical circulation features (e.g. Gill, 1980; Salby and Garcia, 1987; Garcia and Salby, 1987)

87        The NMF decomposition derived by Kasahara and Puri (1981) uses the $\sigma$ coordinates and a realistic vertical temperature

and stability stratification. 3D wave solutions of primitive equations linearized around the state of rest are represented as a
truncated time serie of the Hough harmonic oscillations and the vertical structure functions. The expansion of a global input
data vector $\mathbf{X}(\lambda,\varphi,\sigma) = (u,v,h)^T$ can be represented by a discrete finite series as:

$$\begin{vmatrix} u(\lambda,\varphi,\sigma) \\ v(\lambda,\varphi,\sigma) \\ h(\lambda,\varphi,\sigma) \end{vmatrix} = \sum_{m=1}^{M} \mathbf{S}_m \left[ \sum_{n=1}^{R} \sum_{k=-K}^{K} \chi_n^k(m) \mathbf{H}_n^k(\lambda,\varphi;m) \right] G_m(\sigma) \tag{2}$$

92        The zonal and vertical truncations ($K$ and $M$, respectively) define maximum numbers of zonal waves at a single latitude

(wavenumber $k$) and a maximal number of vertical modes $denoted m$ respectively. Parameter $R$ is the total number of merid-
ional modes which is a sum of the eastward inertio-gravity waves (EIG), westward inertio-gravity waves (WIG) and Rossby
waves. Oscillations in the horizontal plane are given in terms of Hough harmonic functions, $\mathbf{H}_n^k(\lambda,\varphi)$ for every vertical struc-
ture eigenfunctions $G_m(\sigma)$. The horizontal and vertical solutions are connected by the equivalent depth parameter $D_m$, which
appears in Eq. (2) in the diagonal matrix $\mathbf{S}_m$ with elements $(gD_m)^{1/2}$, $(gD_m)^{1/2}$ and $D_m$. Further details of the applied NMF
representation are given in Žagar et al. (2015).

99        The input data vector contains wind components $u,v$ and the geopotential height $h$ defined as $h = g^{-1}P$ where $g$ is the

gravity and $P$ is a modified geopotential given by: $P = \Phi + RT_0 \ln(p_s)$, i.e. the sum of the geopotential field $\Phi$ and a surface
pressure $p_s$ term. Other two variables represent the specific gas constant for dry air ($R$) and the globally-averaged vertical
temperature profile ($T_0$). The nondimensional complex expansion coefficients $\chi_n^k(m)$ represent both geopotential height and
wind perturbations due to waves. The Kelvin mode is represented in (2) by the first eastward-propagating IG mode. Although
our meridional index starts from 1 (to follow otherwise used notation), we shall denote KW in the reminder of this study as the
$n = 0$ EIG mode, i.e. the KWs are given by coefficients $\chi_{kw} = \chi_0^k(m)$.

106       In our application to the L91 ECMWF dataset, we used data on the N64 Gaussian grid and 91 model levels with model

top located at 0.01 hPa (around 80 km). Data are analyzed 4 times per day, at 00, 06, 12 and 18 UTC. The pre-processing
step consists of the interpolation of winds and geopotential from the hybrid ($\sigma - p$) levels to $\sigma$ levels after geopotential $\Phi$ is



computed on the hybrid levels. The truncation values are $K = 55$ and $M = 60$. Higher vertical modes were left out as their
contribution is negligible in the outputs in the TTL and the stratosphere. The relation between the truncation parameters and
the normal-mode projection quality is discussed in Žagar et al. (2015) and references therein.
Once the forward projection is carried out and coefficients $\chi_n^k(m)$ are produced, filtering of KWs in physical space can be
performed through (2) after setting all $\chi$, except those representing the KWs, to zero. The result of filtering are fields $u_{kw}$,
$v_{kw}$ and $h_{kw}$ which provide the KW zonal wind, meridional wind and geopotential height perturbations. Notice here that in
contrast to the equatorial $\beta-$plane, KWs on the sphere have a very small meridional wind component which is thus left out
from the discussion.
The KW temperature perturbation, $T_{kw}$ can be derived from the $h_{kw}$ fields on $\sigma$ levels using the hydrostatic relation in $\sigma$
coordinates:
$$T_{kw} = -\frac{g\sigma}{R}\frac{\partial h_{kw}}{\partial \sigma}.$$      (3)
The orthogonality of the normal-mode basis functions provides KW energy as a function of the zonal wavenumber and
vertical mode. After the forward projection, the energy spectrum of total (potential and kinetic) energy for each Kelvin wave
can be computed using the energy product for the $k$th and $m$th normal modes (Žagar et al., 2015) as:
$$I_{\mathrm{kw}}(k,m) = \frac{1}{2}gD_m\,\chi_{kw}[\chi_{kw}]^*.$$      (4)
The units are J kg$^{-1}$. The KW global energy spectrum as a function of the zonal wavenumber is obtained by summing energy
in all vertical modes:
$$I_{\mathrm{kw}}(k) = \frac{1}{2}\sum_{m=1}^{M} gD_m\chi_{kw}[\chi_{kw}]^*.$$      (5)
**2.2  Examples of 3D structure of Kelvin waves in L91 analyses**
Kelvin waves are shown in Fig. 1-2 for a few days in July 2010 to introduce and illustrate their properties as filtered by the
NMF methodology. The second part of July 2010 was characterized by an abundancy in both vertically propagating as well as
quasi-stationary KW structures throughout the atmosphere.
Figure 1 illustrates the meridional structure of Kelvin waves on 25 July 2010 on 2 levels. KW activity was found largest in
the zonal wind component at 150 hPa over the Indian Ocean. The geopotential dipole structure is centered over the convective
hotspot over the Maritime continent. At 100 hPa, we find largest amplitude of KW temperature perturbations up to 4 K
positioned above the zonal wind maxima at 150 hPa. The meridional wind component of the KW is at most 0.22 ms$^{-1}$ at 100
hPa which is negligible compared to the zonal wind component (maximum 12.5 ms$^{-1}$) making the KW wind field primarily
zonal. Note that the presented horizontal structure at a single level is a superposition of 60 vertical modes, i.e. 60 shallow water
models with equivalent depts from about 10 km to a couple of meters.
Figure 2 can be discussed in relation to Eq. (3). It states that the amplitude of the cold (warm) KW temperature perturbation
is proportional to the negative (positive) vertical gradient in geopotential, as well as in zonal wind since zonal wind and





geopotential components are in phase. Horizontally, the cold anomaly is always located between the westerly and the easterly
phase of the zonal wave component. Vertically, maximum positive temperatures are observed between easterly winds below and
westerly winds above. A rough estimation can be made of the vertical wavelength based on alternating zonal wind minima and
maxima. For example, on 31 July a quasi-stationary vertical wave structure with extension in the stratosphere located around
60°E has easterly winds located at 50 hPa ($\sim$ 21.5 km) and 150 hPa ($\sim$ 13.5 km), which makes a vertical wavelength of around
8 km. In the stratosphere, above 80 hPa, strong KW activity was present in the form of free waves propagating eastward and
downward, therefore with upward transport of KW energy (Andrews et al., 1987). KW amplitudes were largest over Eastern
hemisphere with temperatures up to 4 K and zonal winds up to 12 ms$^{-1}$. The large amount of KW activity occurred during
the easterly phase of the QBO with strong easterly winds present between 30 and 80 hPa (not shown), providing favourable
conditions for the waves to propagate upward.
Between 100 and 200 hPa during the second half of July, there was low-frequency KW activity present in the form of a
stationary and robust "wave-1" pattern with strong KW easterly winds up to 24 ms$^{-1}$ in Eastern Hemisphere and KW westerly
winds up to 10 ms$^{-1}$ in Western Hemisphere. The high vertical resolution within the TTL resolves shallow KW structures and
a typical slanted structure towards the east in KW easterlies as well. The appearance and strength of horizontal KW response
coincides with the presence of strong easterly winds in the TTL in the Eastern Hemisphere during this period (not shown).
Figure 2 also shows that below 300 hPa the KW activity decreases and we shall not discuss levels under 300 hPa in the
paper. More examples based on daily basis filtered from the 10-day deterministic forecast of the ECMWF can be found on the
MODES website[1].
**2.3  Other data and impact of the background state**
In addition to the outputs from modal decomposition, full zonal wind and temperature fields from ECMWF analyses are used
to compute the background fields based on the same N64 grid and over the same period (Jan 2007 - Jun 2013). Zonal wind $U$
and static stability $N$ are latitudinally averaged in the belt 5°S-5°N on all model levels to produce their zonal structure.
Static stability profiles are estimated through
$$N^2 = \frac{g^2}{\Theta}\frac{\partial \Theta}{\partial \phi} \tag{6}$$

in units of s$^{-2}$ and are defined on hybrid model levels on which the geopotential field $\phi$ and the potential temperature field $\Theta$
are derived a priori from the input data. Both fields are shown in Fig. 3.
The zonal wind field has the largest values on average in the TTL around 150 hPa with westerly winds peaking in the
Western Hemisphere over the Pacific Ocean and easterly winds peaking in the Eastern Hemisphere over the Indian Ocean
and Indonesia. It represents a typical time-averaged outflow pattern in response to tropical convection (e.g. Fueglistaler et al.,
2009). Throughout the seasons there is a longitudinal shift of this pattern following the convective source which is most clearly
observed at 150 hPa. Such seasonal shift is visible up to 100 hPa in Fig. 3(b) where winds are weaker compared to 150 hPa.
In northern winter, zonal winds are strongest over Indonesia and Eastern Pacific with the zonal wind maxima position and

---

[1]http://meteo.fmf.uni-lj.si/MODES/





strength similar compared to the longer ERA-40 dataset used by Suzuki and Shiotani (2008). During northern summer easterly
winds mainly prevail over the Indian Ocean, which is linked to the Indian Monsoon season.
At 100 hPa, the static stability illustrates the strongest seasonal cycle with values ranging from near-tropospheric values of
$3 \times 10^{-4}$ ms$^{-2}$ during northern winter towards stratospheric values of $5 - 6 \times 10^{-4}$ ms$^{-2}$ during northern summer. Note also
the resolved local maxima in static stability at 80 hPa above the warm pools, known as the Tropical Inversion Layer (TIL) and
which is possibly wave-driven (Grise et al., 2010; Kedzierski et al., 2016). Figure 3(b) suggests that the TIL descends down to
100 hPa during the summer months peaking over Western Pacific, in agreement with the cycle found in GPS-RO observations
by Grise et al. (2010).
Kelvin waves are subject to wave modulation in changing background environments. Along its trajectory, the potential
energy of the KW changes with varying background winds and stability which can be largely described by linear wave theory
as long as waves are not near their critical level involving breaking and dissipation (Andrews et al., 1987). For simplification,
KW modulation can be examined for the case of pure zonal as well as pure vertical wave propagation based on the wave
modulation analysis performed by Ryu et al. (2008). A few key points on their local wave action conservation principle are
summarized in the following.
In the tropical atmosphere, zonal modulation is the dominant process for KWs propagating in the stratosphere and in all non-
easterly winds in the TTL. Vertical modulation becomes important in the presence of easterly winds within the TTL. Zonal
modulation is found to affect both $u_{kw}$ and $T_{kw}$ components and their amplitudes are proportional to the Doppler-shifted phase
speed by $(c - U)^{1/2}$ in case of pure zonal propagation direction. This means that Kelvin waves diminish in amplitude over
regions with westerly winds and become more prone to dissipative processes, while amplify over regions with easterly winds[2].
In case of pure vertical modulation, the change in wave potential energy mainly resonates with the temperature component of
the Kelvin wave. Along the rays' vertical path, the waves amplitude is proportional to the Brunt-Väisälä frequency as $\propto N^{3/2}$,
and to the Doppler-shifted phase speed as $\propto (c - U)^{-1/2}$, such that $N$ is expected to play a primary role above 120 hPa where
its value starts increasing rapidly (see Fig. 3).
Alexander and Ortland (2010) showed through wave modulation principles that temporal variations in zonal-mean $N$ indeed
are correlated with observed KW amplitudes at 16 km (approx. 100 hPa). A more extensive wave modulation analysis was
described by Flannaghan and Fueglistaler (2013) using the full ray tracing equations to demonstrate that zonal winds in the TTL
not only modulate Kelvin waves locally, but also create a lasting modulating effect on wave activity through ray convergence
in the stratosphere. In particular, the seasonal cycle of the upper tropospheric easterlies (on average located over the western
Pacific), that acts as an escape window for Kelvin waves throughout the year and largely explains the longitudinal structure of
Kelvin wave zonal wind and temperature climatology.
We shall present the seasonal variability of tropical convection by using the Outgoing Longwave Radiation (OLR) dataset
with daily outputs from the NOAA Interpolated OLR product (Liebmann and Smith, 1996). The OLR product, often used as
a proxy for convection, is extracted on a $2.5° \times 2.5°$ grid and interpolated on a N64 grid. Latitudal averages are derived over

---

[2]Keeping in mind that vertical wave propagation and consequently modulation becomes increasingly important as well wherever easterly winds are strong.



larger domain, namely over 15°S-15°N since organized convection tend to happen more remote from the equator, especially
during the summer monsoon season over the Asian continent.

## 3  Kelvin wave energetics

We start with an overview of KW energy distribution among the zonal wavenumbers as given by (5), followed by the seasonal
cycle of KW energy as a function of zonal wavenumber.

### 3.1  Energy distribution of Kelvin wave

The seasonal cycle in the energy-zonal wavenumber spectra is shown in Fig. 4 after summing up over all vertical modes. On
average, energy decreases as the zonal wavenumber increases as typical for atmospheric energy spectra. As we deal with the
large scales, we show only the first six zonal wavenumbers with energy values shown separately for the annual mean and the
four seasons separately.
Figure 4 shows that largest seasonal variations in KW energy are found at the largest zonal scales. For all zonal wavenum-
bers, above annual-mean energy values are observed during winter and summer seasons while autumn and spring are below
annual-mean energy. In the zonal wavenumber 1, total KW energy varies between 200 Jkg$^{-1}$ in MAM season and somewhat
over 300 Jkg$^{-1}$ in JJA. In wavenumber 2, values do not exceed 100 Jkg$^{-1}$ and JJA still contains the largest energy. At higher
wavenumbers, DJF season becomes the most energetic. In $k > 4$, total KW energy is under 20 Jkg$^{-1}$ and continue to reduce
with $k$. The slope of the KW energy spectrum is between $-5/3$ and $-1$ at planetary scales (not shown), similar to the spectra
presented in Žagar et al. (2009) for July 2007 data. The summer spectra has on average the steepest slope compared to other
seasons, in particular the winter spectra. The energy distribution on planetary scales is mainly associated with large-scale trop-
ical circulation established in response to ongoing tropical convection. Therefore, the zonal distribution of tropical convection
may likely play a crucial role in explaining winter and summer season differences of KW energy, which will be explored in
next section.

### 3.2  Seasonal cycle of KW energy

Figure 5 illustrates more details on the seasonal cycle by showing KW energy time series at the largest scales represented by
zonal wavenumbers $k = 1$, $k = 2$ and remaining scales $k > 2$. During most summers and occasionally in winter (e.g. 2008)
the total amount of KW energy in $k = 1$ can reach up to 600 Jkg$^{-1}$, or twice the summer average. The minimum in $k = 1$
KW energy mainly occurs during October month followed by April with values dropping towards 100 Jkg$^{-1}$, or half the
autumn average. The temporal pattern in $k = 2$ is similar to the $k = 1$ pattern, but with a less pronounced semiannual cycle
with maximum values up to 200 Jkg$^{-1}$ and minimum values towards 30 Jkg$^{-1}$. On zonal scales $k > 2$, KWs still show a
semiannual cycle with highest vertically-integrated values of energy over winter seasons.
In particular, for zonal wavenumber $k = 1$ one can distinguish inter-monthly in addition to semiannual variability. Inter-
monthly variability is most clearly observed during northern summer, for example in July 2011 where one can distinguish



six separate peaks of over $400$ $\mathrm{Jkg}^{-1}$ energy over a period of approximately 90 days resembling an average wave period of about 18 days. These are typical periods for free propagating Kelvin waves as observed in the TTL and lower stratosphere (e.g. Randel and Wu, 2005). Note here again that our KW energy is vertically integrated over the whole model depth. This means that the observed intermonthly variability of KWs appears dominated by the cyclic process of free propagating KWs entering the TTL, amplifying due to changing environmental conditions, followed by wave breaking or dissipation.

The dominant scales of temporal variability in KWs are illustrated by a frequency spectrum of $k = 1$ in Fig. 6. The spectrum is produced by a Fourier transform of energy data time serie of 6.5 years to frequency space. The resulting power spectrum has been smoothed by taking the Gaussian-shaped moving averages over the raw spectrum by using a Daniell kernel three times (Shumway and Stoffer, 2010). The spectrum shows a clear peak at 1-day period representing tidal variability in KWs. After that, a gradual increase of energy is seen towards the 16-day period with multiple individual periods standing out. For periods longer than 20 days, individual peaks are found close to 25, 43 and 59 days. After that, most KW energy is contained by far in the semiannual cycle. The frequency spectrum provides an useful starting point for the discussion in the next section when the spatiotemporal patterns of KWs shall be examined in several spectral domains.

Returning to Fig. 5, a low-pass filter with 90 day cut-off has been applied on KW energy in order to keep only the two main spectral peaks in Fig. 6. The result is visible as the thicker black line in Fig.5 for all three zonal wavenumber groups. A semiannual cycle for all zonal wavenumbers is evident with most energy observed around January and July, while least energy is observed approximately one month after the equinoxes. During the years 2007, 2010, 2011, and 2012, more $k = 1$ KW energy is observed during summer compared to the follow-up winter. The winter of 2009-2010 was for example above average with energy values for $k = 1$ above 350 $\mathrm{Jkg}^{-1}$.

The year to year differences can be explained by many coupled factors: In general, one expects vertically-integrated KW activity to increase when background wind conditions become favorable, i.e. in the presence of easterly winds. This occurs in the TTL in relation to strong convective outflow (Garcia and Salby, 1987; Suzuki and Shiotani, 2008; Ryu et al., 2008; Flannaghan and Fueglistaler, 2013) during winter and summer seasons mainly. Moreover, one can expect enhanced KW activity whenever the easterly QBO cycle is present in the stratosphere (Baldwin and Coauthors, 2001; Alexander and Ortland, 2010) or when the ENSO index is positive (Yang and Hoskins, 2013). The latter factor might explain partly the large difference in the abundant amount of KW energy during the El Niño winter of 2009-2010 and the below-average amount of KW energy a year after during the strong La Niña winter of 2010-2011. However, during the La Niña winter of 2007-2008, the amount of KW energy is observed to be above normal. That winter was however characterized by favorable easterly QBO conditions in the stratosphere while during the winter of 2010-2011 stratospheric winds were largely westerly of nature thereby prohibiting KW activity. The role of these low-frequency atmospheric phenomena on KW seasonal variability is a topic of further research.

Finally, Fig. 5 also shows that July 2007, previously examined by Žagar et al. (2009), was an exceptionally energetic month. A large part of that energy, approximately 400 $\mathrm{Jkg}^{-1}$ (52.7% of total KW energy), was projected on zonal wavenumber 1. In spatiotemporal terms, it represented the presence of a strong dipole structure in the TTL (as in Fig.2), which is colocated with favourable easterly wind conditions in the TTL as well as in the stratosphere (not shown).



## 4  A spatiotemporal view on Kelvin wave seasonal variability

### 4.1  Kelvin wave decomposition among wave periods

In this section, the spatiotemporal view of KWs shall be presented over three dominant ranges of wave periods in Fig. 6, namely: (i) the (semi)annual cycle using a low-pass filter with cut-off period at 90 days, (ii) the intraseasonal period using a bandpass filter over periods between 20-90 days, and finally (iii) the intramonthly period with bandpass filtered periods between 3-20 days. The choice of ranges, especially the intramonthly periods is related to previous studies using observations. For all three cases, mean 6 year fields as well as seasonal means shall be presented.

Both KW components $u_{kw}$ and $T_{kw}$ are Fourier-transformed to frequency space where the spectral expansion coefficients $\chi_{kw}$ in domains outside the desired frequency ranges are put to zero. Case (i) results in KW components $u_{kw,l}$ and $T_{kw,l}$ where $l$ indicates the low-frequency component. Case (ii) results in $u_{kw,m}$ and $T_{kw,m}$ where $m$ indicates the intramonthly period. Case (iii) results in fields $u_{kw,h}$ and $T_{kw,h}$ where $h$ stands for the high-frequency component. Previous studies have defined free propagating Kelvin waves over similar ranges (3-20 days, Alexander and Ortland (2010); 4-23 days, Suzuki and Shiotani (2008)) and similarly for intraseasonal periods (23-92 days, Suzuki and Shiotani (2008)). Next, seasonal averages will be taken over the four seasons, resulting in variables $\overline{u_{kw,l}}^{s}$, $\overline{T_{kw,l}}^{s}$ for the low-frequency component and similarly for the other two cases. The superscript $s$ represents one of the four seasons: northern winter ($s =$ DJF), spring ($s =$ MAM), summer ($s =$ JJA), and autumn ($s =$ SON).

Cases (ii) and (iii) contain purely subseasonal variability and therefore one can expect their mean 6-year fields to be zero-valued since variability beyond 90 days has been put to zero. Similarly, mean fields for each of the four seasons results in $\overline{u_{kw,h}}^{s} \ll \overline{u_{kw,l}}^{s}$ and $\overline{u_{kw,m}}^{s} \ll \overline{u_{kw,l}}^{s}$ and the same for the temperature component. This reflects the fact that positive and negative phases of the fast KW responses average out to approximately zero on seasonal timescales (figure not shown). Therefore, the seasonal mean over the absolute amplitudes for zonal wind and temperature are examined instead, i.e. $\overline{|u_{kw,h}|}^{s}$, $\overline{|u_{kw,m}|}^{s}$ and similarly for temperature component, in order to study seasonal fluctuations in subseasonal KW amplitudes.

Figure 7 shows results for all three cases after taking mean over the whole period. The left panel resembles a dominant "wave-1" structure with zonal wind maximized around 140 hPa. Easterly KW winds are strongest around $60°$E and westerly winds around the Date Line. Note that two stationary perturbations over African ($30°$E) and South American ($80°$W) orography are the result of our terrain-following NMF analysis. If one compares the KW zonal wind pattern with the climatological zonal wind pattern in Fig. 3(a) it can be observed that the zonal wind pattern is located around $20°$ west of the climatological pattern. Wave temperature perturbations are largest where the vertical gradients in zonal wind are largest which explains the quadripole structure. Heating (cooling) by KWs is located at 100 hPa in Eastern (Western) Hemisphere and the other way around at 200-300 hPa.

The middle panel of Fig. 7 shows the average distribution of KW activity on intraseasonal timescales. The activity is largest in the Eastern Hemisphere with average zonal wind maxima up to 3 ms$^{-1}$ and temperature maxima up to 0.7 K. Zonal wind activity is largest over a broad area between 90 and 150 hPa over the Indian Ocean and the Maritime Continent. Temperature



activity occurs slightly higher around 90-100 hPa. Intraseasonal activity is locally somewhat increased also around $120°$W,
west of the Andes mountain range.
Finally, Fig. 7c illustrates the average distribution of free propagating KWs. The Eastern Hemisphere again makes up for the
larger KW activity than the Western hemisphere, but the maximum is located more upward in comparison to the intraseasonal
scales, around 80 hPa. Zonal wind activity peaks up to 3 ms$^{-1}$ over a broad range of 70-110 hPa and temperature peaks over
a more narrow area around 76 hPa (up to 0.75 K). The main area for KW activity is found over Indian Ocean region, while
least wave activity is above central Pacific. Towards the stratosphere KW activity reduces and becomes more uniform along in
longitudal direction.

## 4.2  Low-frequency Kelvin wave variability

The seasonal patterns of the low-frequency components of the KW (from hereon referred to as the Gill-type KW response) is
presented as pressure-longitudinal cross-sections along the equator (at $0.7°$N) of the KW seasonal means, given by $\overline{[u_{kw,l}]}^s$
and $\overline{[T_{kw,l}]}^s$ in Fig. 8.
The largest Gill-type KW response is found during NH summer. A strong dipole "wave-1" pattern is evident in the TTL. The
strongest zonal winds are found close to 150 hPa with easterlies up to -12 ms$^{-1}$ centered over Indian Ocean and westerlies
up to 6 ms$^{-1}$ over the Western Pacific. Negative temperature KW anomalies at 110 hPa are strongest as well during JJA with
values up to 1.5 K over Indian Ocean and annually averaged value of -0.5 K over Western Pacific.
During NH winter, the dipole pattern is shifted more eastward and upward compared to NH summer and has a more slanted
structure. Easterly (westerly) KW winds are located more east over the Maritime continent (central Pacific) and centered at 130
hPa. The upper temperature dipole pattern is found higher up at 90 hPa approximately. Values are somewhat weaker compared
to NH summer with easterlies up to -6 ms$^{-1}$ and westerlies up to 5 ms$^{-1}$.
Finally, NH autumn and spring seasons are transition seasons with respect to the strength and position of the KW dipole as
it moves west- and downward towards summer and east- and upward towards winter. NH spring has the weakest KW dipole
with slightly stronger westerly winds up to 5 ms$^{-1}$.
The longitudinal position and the strength of the Gill-type KWs have been linked to the seasonal patterns of the background
winds in the TTL representing the upper level monsoon and Walker circulations (Flannaghan and Fueglistaler, 2013). The
average background winds maximize at 150 hPa as shown in Fig. 3(a). In Fig. 8, one can see how the KW easterlies in
Eastern Hemisphere are strongest during NH summer in relation to the Indian-South Asian monsoon circulation. Background
easterlies as strong as -30 ms$^{-1}$ are located approximately $10°$ east of the KW maximum easterlies. NH winter has the strongest
background westerlies in relation to the upper-level circulation of the Western Pacific anticyclones. NH spring (autumn) shows
similar background wind patterns compared to NH winter (summer) but with weaker circulation.
Further details on longitudinal position and interannual variability of Gill-type KW response at its maximum value at 150
hPa are illustrated by the Hovmoller diagram in Fig. 9. For comparison, tropical convection is represented as well through
the OLR proxy variable averaged over $15°$S-$15°$N latitudes. All fields have been filtered with a 90 day cut-off low-pass filter
in order to highlight the seasonality. As a result, one can observe enhanced/reduced Gill-type KW activity during the same





individual seasons as seen from the timeseries in Fig. 5. Above average seasonal KW activity with stronger Gill-type structures
occurred during the summer of 2007 (mainly through its easterlies at $60°$E) and during the winters of 2006-2007 and 2009-
2010. In these winters, El-Nino was active and a clear longitudinal eastward shift is observed in OLR, in the background
circulation (not shown), as well as in the Gill-type KW structure. The El-Nino winter of 2009-2010 was followed by a strong
La Nina winter with an increase in tropical convection over the Maritime continent (note: OLR values below 195 Wm$^{-2}$).
The vertical seasonal movement of the KW dipole has been linked with the seasonal movement of the tropical tropopause
height (Flannaghan and Fueglistaler, 2013; Ryu et al., 2008). The position of the tropical tropopause height (represented by
a static stability value of $5 \times 10^{-4}$ s$^{-2}$ in Fig. 8) is found at approximately 85 hPa during winter and descends towards 100
hPa in summer, similar to values obtained from GPS-RO observations by Grise et al. (2010). In particular, during summer,
one can notice how the asymmetry in the tropical tropopause height over Indian Ocean around $60°$E coincides with increasing
temperatures by the KW dipole up to 1.5 K. Such deformation of the tropical tropopause is also evident during winter and
autumn seasons.
Figures 10a and 10b illustrate seasonal-mean KW temperatures $\overline{T_{kw,l}}^s$ in relation to the tropical tropopause layer defined
by static stability $N^2$. Seasonal variations in KW temperatures are colocated with the position of the tropopause, descending
down from its highest position during winter to its lowest position during summer. Temperature amplitudes are observed to
decline roughly above $N^2 = 5 - 6 \times 10^{-4}$ s$^{-2}$. Within this zonal-mean seasonal picture, zonal asymmetries in $N^2$ exist and
are found: (i) near the Date Line with values of $8 \times 10^{-4}$ s$^{-2}$ at 80 hPa during winter and $7 \times 10^{-4}$ s$^{-2}$ at 90 hPa during
summer and (ii) lower at 100 hPa over the Indian Ocean during summer. Particularly during NH summer, the deformation
of the zonal-mean static stability field colocates strongly with the position of a strong KW temperature anomaly over Indian
Ocean. A rough estimation is made on the contribution of the KW anomaly to the zonal deformation of the tropopause layer by
removing zonal-mean parts of both fields. First, static stability zonal anomalies, $\overline{N'^2}^s$, are derived by subtracting zonal-mean
values of $N^2$ from the full $N^2$ field per timestep and at every pressure level, followed by seasonal averaging. Next, we can
estimate the static stability change associated with the KW anomaly, using the relation: $N_{kw}^2 = \frac{g}{\theta}\frac{\partial \theta_{kw}}{\partial z}$, followed by seasonal
averaging as well, i.e. $\overline{N_{kw}^2}^s$.
As a result, Fig. 10c and 10d show how both static stability anomalies are overlapping. During winter, the structure of the
zonal anomaly $\overline{N'^2}^s$ has a positively-valued tilt eastward which stretches up to 80 hPa, while during summer a strong static
stability anomaly is found more localized over Indian ocean region with values in the TTL up to $\overline{N'^2}^{JJA} = \pm 0.8 \times 10^{-4}$ s$^{-2}$.
The anomaly associated with the KW temperature anomaly is found to peak up to $+0.6 \times 10^{-4}$ s$^{-2}$ during summer and up
to $+0.4 \times 10^{-4}$ s$^{-2}$ during winter. Finally, by dividing both fields with each other, the resulting contribution of the quasi-
stationary Kelvin wave to the observed deformation of the tropical tropopause layer is estimated up to 60% (80%) during NH
summer (winter).
**4.3 Intraseasonal Kelvin wave variability**
The seasonality of intraseasonal Kelvin wave variability is shown in Fig. 11 and shall be briefly discussed here. The NH
winter stands out as the most active season for KW activity, located mainly in the Eastern hemisphere centered at $100°$E and





with maximum activity at 110 hPa for zonal wind and temperature with a second maximum in temperature at 90 hPa. Values
observed are up to 0.8 K for KW temperature and 5 ms$^{-1}$ for KW zonal wind. During NH spring season, the KW activity fields
are weaker but spread over a larger area in the Eastern hemisphere and in the TTL with maximum activity centered at 120 hPa
(90 hPa) for the zonal wind (temperature) component. Both NH summer and autumn seasons have KW activity positioned
at lower altitudes and more westward. In both seasons, KW zonal wind activity is split up between two structures with an
eastward tilt with height; one with a maximum around 110°E and one pattern starting from 100 hPa and extending towards
60°E. Note also the increase in KW activity in the Western hemisphere below 150 hPa in the East Pacific. The maximum KW
activity in the temperature component for both seasons is positioned near 100 hPa approximately on the tropical tropopause
contour with value $5 \times 10^{-4}$ s$^{-2}$.
The eastward tilted structure is observed throughout all seasons except NH spring when background easterly winds are nearly
absent in the Eastern hemisphere. In all other seasons one can observe how the tilted structure is locked to the background
easterlies with maximum amplitudes located slightly above and west of it. Such eastward tilt with height has been frequently
observed, for example over radiosonde station Medan at 100°E during the early stage of MJO development (Kiladis et al.,

384  2005).

**4.4    Free propagating Kelvin waves**
The seasonal variability of free traveling Kelvin waves, represented by their absolute amplitudes $\overline{|u'_{kw,h}|}^{s}$ and $\overline{|T'_{kw,h}|}^{s}$, shall
be examined in relation to the background conditions. Figure 12 illustrates favorable regions for KW activity. In general, KW
activity increases upward from around 120 hPa towards its zonal-mean peak value at 76 hPa. The largest values are observed
in EH in region from 30°E till 150°E. The temperature component in particular has a constant maximum peak (up to 0.8 K
in EH) located around 76 hPa throughout the year, where also the largest increase in $N^2$ occurs as shown in Fig. 3. Above 70
hPa, KW activity continuously decreases in the stratosphere.
The longitudinal structure of the KW zonal wind shows two distinct peaks in the TTL, one consistently located at 76 hPa
and another around 100-110 hPa in the EH which is mainly present during solstice seasons. The first maximum coincides
with the temperature distribution which can be explained by their balance relationships and free horizontal propagation in the
stratosphere. Below the tropopause, KW activity is coupled to convective processes alternating the tropospheric vertical wave
structures as discussed by Flannaghan and Fueglistaler (2012).
The secondary maximum around 110 hPa in Fig. 12 is present mainly during solstice seasons in EH and it is associated
with the seasonal movement of the background wind. The maximum of KW wind and the background wind maximum move
eastward from winter to summer season similar to the low-frequency variability. A day-by-day comparison of the KW activity
and background wind confirms that propagating KWs amplify while approaching a region of strong easterlies, forming a
folding structure around it while the individual KWs dissipate towards the center of easterly winds. One can notice in Fig. 12 a
fast reduction of KW amplitudes eastward of its maximum towards the center of the background easterlies. It is likely related
to dissipation and wave breaking processes as observed over Indonesia (120°E) by Fujiwara et al. (2003). Within such regions,



the KW-background wind interaction becomes complex and the linearity assumption breaks (Ryu et al., 2008; Flannaghan and
Fueglistaler, 2013).
A comparison with the previous study by Suzuki and Shiotani (2008) using ERA-40 data shows that the L91 data contain
stronger KW activity in the vicinity of the background easterlies in the Eastern Hemisphere, and more fine-scale details which
can be explained by better analyses based on more observations and improved models including increased resolution. For
example, Suzuki and Shiotani (2008) used 5 levels of ERA-40 data between 50 and 200 hPa whereas the present study considers
25 model levels between $50 - 200$ hPa. Maxima of the KW temperature signal appear in similar locations and strength except
for a small offset in vertical position (70 hPa in Suzuki and Shiotani (2008) versus 80 hPa in Fig. 12) and a larger zonal
asymmetry in our results.
Another view of the seasonal cycle of free propagating KWs is illustrated in Fig. 13 which focuses on the spatiotemporal
distribution of individual KW packets. Hovmoller diagrams of KW zonal wind and temperature at levels 110 and 200 hPa from
different years are shown along with the background zonal wind. In addition, the monthly-mean values of daily maximum KW
amplitudes occurring at a specific longitude along the equator are added next to each diagram.
The individual wave tracks at 110 hPa illustrate KWs with amplitudes exceeding 3 ms$^{-1}$ and 0.6 K which are propagating
throughout the year in the Eastern Hemisphere, during June-October months only over the Pacific, and all except winter
months in most of the Western Hemisphere. Typical wave tracks start east of the 0° (30°W) meridian during winter (summer)
and largely disappear west of 120°E. The largest wave amplitudes are observed between 50°E and 100°E prior to regions
of easterly winds in agreement with Fig. 12. Here presented details show that most notable waves appear during the Asian
monsoon period with upper-level easterlies prevailing from June into September. The largest Kelvin wave amplitudes appear
confined to the June and July months followed by a rapid drop in August. In fact, a local minimum in the number of KWs as
well as in wave amplitudes occurs in August before the KW activity increases slightly during autumn.
At 200 hPa, the favorable area for KW propagation shifts to the Western Hemisphere and large KW activity is observed
west of the South American continent throughout the year (west of 80°W) with a westward extension over the Pacific during
northern summer. Another set of wave tracks starts over equaotial South America around 30°W (5°W) and continues till 60°E
(90°E) during northern summer (winter). The seasonal shifts of approximately 30° in KW tracks colocate with similar shifts
in the prevailing TTL winds.
The amplitude of KWs undergoes a clear annual cycle with a small secondary peak present during northern winter, as
represented by the monthly-means of daily maximum amplitudes along the equator on the rightside of Fig. 13. The largest
amplitudes are found at 110 hPa during NH summer with monthly-mean zonal wind (temperature) values up to 8.5 ms$^{-1}$
(1.8 K) in June. During the winter months Kelvin waves amplify more eastward with monthly-mean zonal wind (temperature)
values up to 7.8 ms$^{-1}$ (1.6 K) in December. At 200 hPa, KW amplitudes are on average lower with a yearly-averaged amplitude
reduction around 55% in temperature and 35% in zonal wind. The semiannual cycle in maximum amplitudes remains visible up
till 70 hPa. Above 70 hPa, where KW activity remains large in Eastern Hemisphere (Fig. 12), the semiannual cycle is replaced
by an interannual cycle in line with the dominant impact of the QBO.





## 5 Discussion and Conclusions


We have applied the multivariate decomposition of the ECMWF operational analyses during the period 2007-2013 when
the operational data assimilation was performed on 91 levels. Model-level data were analyzed every 6 hours. The applied
decomposition provides simultaneously the wind components, geopotential height and temperature perturbations of the Kelvin
waves on the terrain-following levels. As the KW meridional wind component is very small it is not discussed. We focused
on the spatiotemporal features of the KW temperature and zonal wind components in four seasons. The Kelvin wave seasonal
cycle in the tropical tropopause layer (TTL) was compared with seasonal variability of Outgoing Longwave Radiation (OLR),
the background wind and stability fields, which are believed to play an important role for the KW variability. Our study of
the seasonal KW variability complements previous studies which applied different methods for the KW filtering and different
datasets. As KW is a normal mode of the global atmosphere, our filtering of the KW using the 3D-orthogonal normal-mode
function decomposition of global data is a useful approach to quantification of the KW variance. The KW is the most energetic
inertio-gravity mode of the global atmosphere (Žagar et al., 2009) and its representation in weather and climate models is
crucial for reliable simulations of the tropics and its impact on global circulation.
We have presented the total energy of the KWs in the L91 data extending between the surface and 1 Pa as a function of the
zonal wavenumber. Zonal wavenumber $k = 1$ contains a largest portion of KW energy in all seasons. Its energy varies between
~300 in JJA in NH spring to over 400 J/kg in NH summer. In $k = 2$ there is 50% less energy than in $k = 1$ but the NH summer
is still the most energetic season. In all greater zonal wavenumbers, DJF season contains most energy.
Frequency spectrum has revealed a semiannual cycle as well as intraseasonal and intramonthly variability. Three ranges of
wave periods were analyzed: 3-20 days, 20-90 days and longer than 90 days. This choice was partly deliberate in order to
compare our results with several previous studies of KW variability. First we demonstrated that the seasonal-mean KW pattern
in the TTL, with (westerly) easterly winds in the (Western) Eastern hemisphere resembles a time-averaged Gill-type "wave-1"
pattern. The quadrature-shaped temperature component represents a thermally adjusted pattern with respect to the zonal wind
component, and contributes to seasonal (cooling) warming above 100 hPa in the (Eastern) Western hemisphere. The largest
KW amplitudes are observed during summer and winter seasons. From boreal summer towards winter, KW perturbation moves
eastward (from Indian Ocean basin towards Maritime Continent) and upward (e.g. zonal wind component moves up from 150
hPa towards 120 hPa). The KW zonal wind amplitude varies between 12 m/s strong easterlies over Indian ocean near 150 hPa
in JJA to 6 m/s over Western Pacific. Over Indian Ocean in JJA, the KW easterlies thus make almost half of the total wind
vector. The associated KW temperature perturbations are from 1.5 K over Indian ocean in JJA to -0.5 K over West Pacific.
The zonal modulation of Kelvin waves is found to be locked with respect to the seasonal movement of convection and the
convective outflow in the TTL. The modulation effect is strongest for Gill-type Kelvin waves during the summer monsoon
season, when strong easterly winds are present at 150 hPa, resulting in the largest KW zonal wind and temperature anomalies,
of which the latter results in deformation of the tropical tropopause over Indian Ocean.
Intraseasonal (periods 20-90 days) activity is strongest in NH winter with maxima up to 0.8 K for KW temperature and
up to 5 m/s for KW zonal wind centred at 120°E. Both temperature and zonal wind activities have eastward tilt with height.



In comparison to previous study by Suzuki and Shiotani (2008) using ERA-40 data, the slanted structure in the present data
continues to extend more upward and eastward which is likely due to the increased number of vertical model levels compared
to ERA-40. The importance of vertical model resolution for the KW wave structure and amplitude was demonstrated in Žagar
et al. (2012) and Podglajen et al. (2014).

476       For periods $3 - 20$ days, the seasonal cycle of KWs is clearly seen in wave amplitude. The largest amplitudes are located

- from a zonal-mean perspective - between 70 and 100 hPa for both zonal wind and temperature as expected for the free-
propagating Kelvin waves but it is modulated by the seasonal movement of the TTL. A major zonal asymmetry was found in
KW activity: around 110 hPa Kelvin wave undergoes amplification mainly in Eastern Hemisphere during the solstice seasons,
while at 200 hPa a secondary region of KW amplification occurs in Western Hemisphere during boreal summer. Free propagat-
ing KWs show largest amplitudes in the vicinity of the strongest easterlies preferably west and above the center of easterlies.
The NMF methodology has made it possible to observe such dynamics on daily basis whenever easterlies are strong in the
TTL. Nearly real-time representation of the KW activity is available on http://modes.fmf.uni-lj.si.

484       In summary, our seasonal variability analysis shows that the background wind in the TTL linked with convective outflows,

play a dominant role in the longitudinal position where zonal modulation of Kelvin waves is preferred, while the tropical
tropopause and its seasonal vertical movement determines the vertical extent of KW modulation processes.
*Acknowledgements.*   This study was funded by the European Research Council (ERC), Grant Agreement no. 280153 MODES, http://meteo.fmf.uni-
lj.si/MODES.





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

**Figure 1.** Horizontal map slices centered on the Tropical Belt on the 25th of July 2010 (panel b in Fig. 1) at (a) 100 hPa and (b) 150 hPa. Kelvin wave wind fields are represented by blue vectors. Contour fields indicate KW geopotential ($h_{kw}$) (black contours, every 20 m) and temperature ($T_{kw}$) (red contours, every 1 K). Dashed contours represent negative values for geopotential and temperature components.

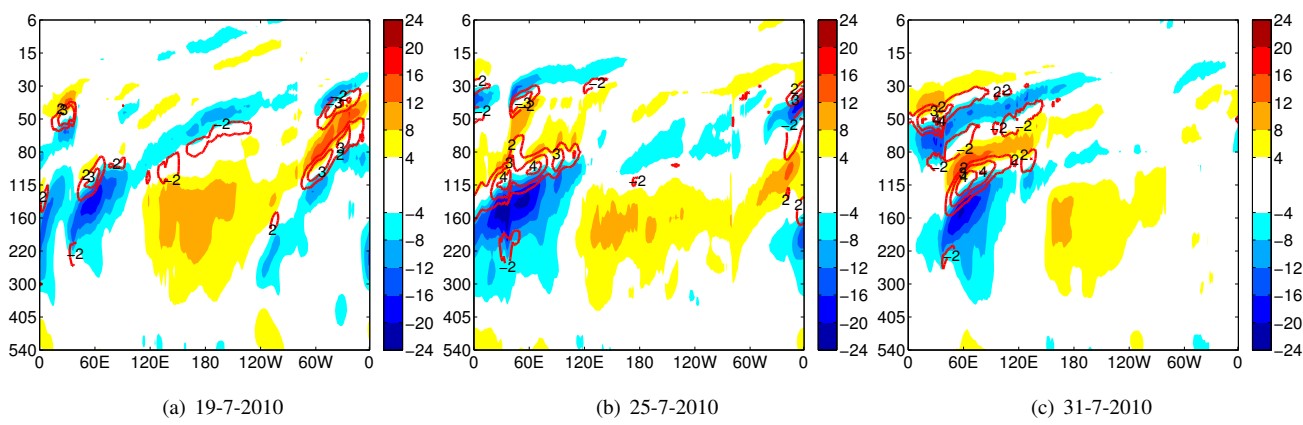

(a) 19-7-2010          (b) 25-7-2010          (c) 31-7-2010

**Figure 2.** Longitude-pressure sections along $0.7°$N of Kelvin wave zonal wind (red-blue shaded contours) and temperature (red contours) on the following days in July 2010: (a) 19, (b) 25 and (c) 31. Temperature is given every 1 K, contours starting at 2 K. Zonal wind values are given every 4 ms$^{-1}$.





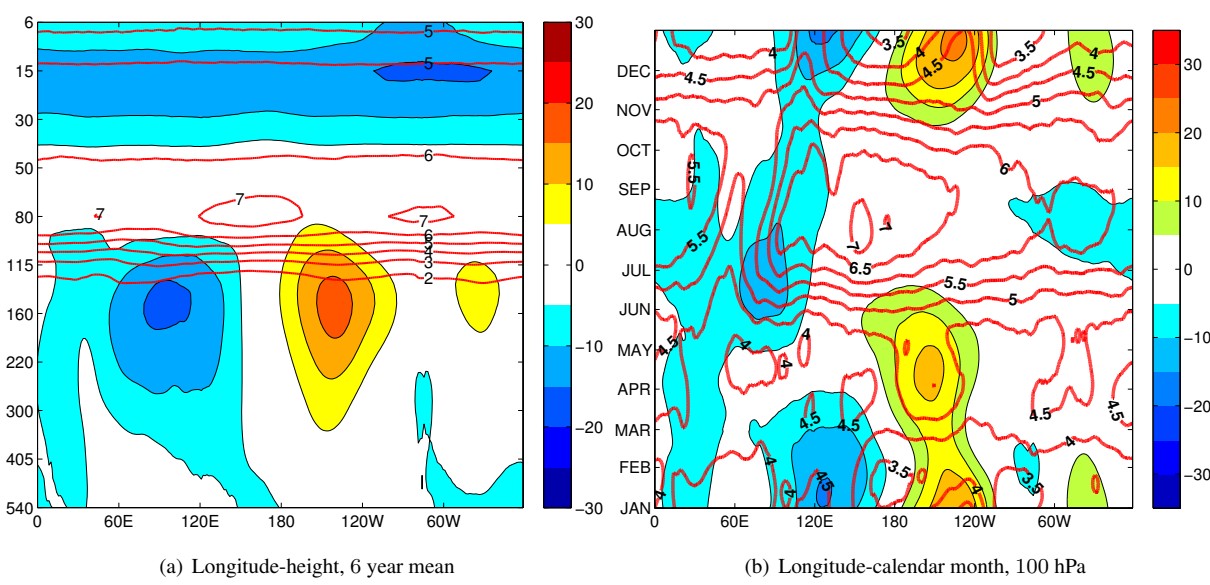

(a) Longitude-height, 6 year mean            (b) Longitude-calendar month, 100 hPa

**Figure 3.** Zonal wind (blue-to-red contour fields, each $5 \text{ ms}^{-1}$) and static stability given by Eq. (6) (red contours each $1 \times 10^{-4} \text{ s}^{-2}$ for a) each $0.5 \times 10^{-4} \text{ s}^{-2}$ for b) panel). (a) Longitude-height section averaged over 6 years and (b) Longitude-time section at 100 hPa. Both fields are latitudinally averaged over $5°$S-$5°$N, and have been low-pass filtered a priori with a cut-off period of 90 days to highlight seasonal variability.





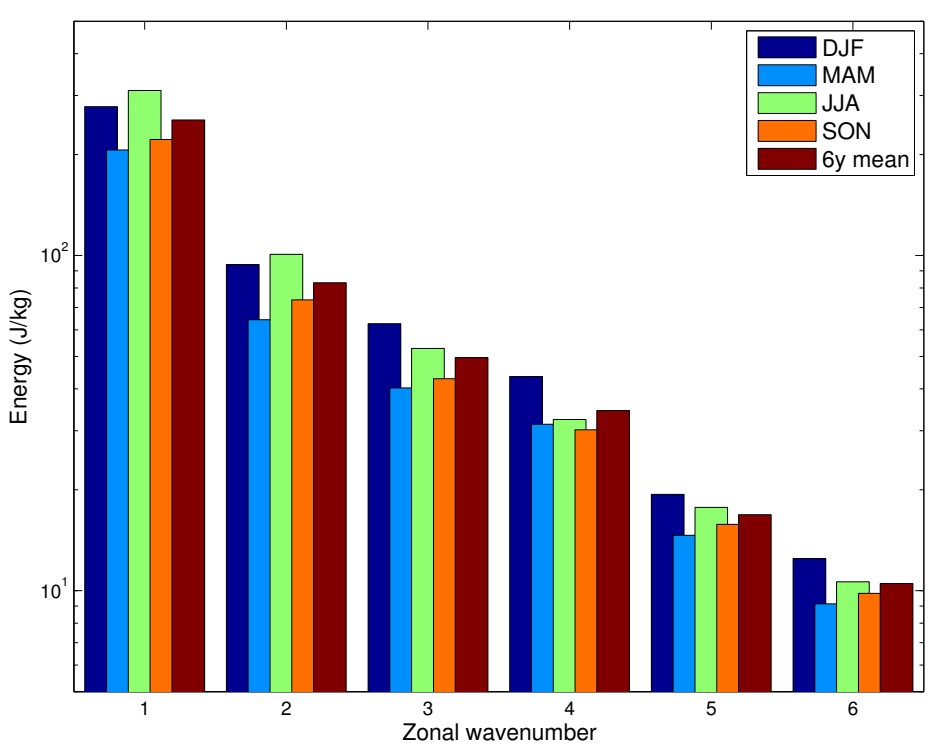

**Figure 4.** Kelvin wave energy (in $Jkg^{-1}$) as function of zonal wavenumbers $k$ for $k = 1 - 6$. For each $k$, energy values are averaged over the following periods from left (blue bars) to right (red bars): DJF (winter), MAM (spring), JJA (summer), SON (autumn) and for the full $6, 5$ year period. Energy is summed over all vertical modes.



(a) $2007 - 2009$

(b) $2010 - 2012$

**Figure 5.** Timeseries of KW energy (in Jkg$^{-1}$) for various zonal wavenumbers over the following periods: (a) $2007 - 2009$ and (b) $2010 - 2012$ (incomplete year 2013 has been left out). 'A', 'J' and 'O' refer to the first day of April, July and October months. Energy values are summed over all vertical modes and for the following zonal wavenumbers: planetary scales $k = 1$ (blue) and $k = 2$ (green) and all smaller zonal scales, $k > 2$ (red). A 90-day low-pass filter has been applied (black lines) for each of the zonal wavenumber groups in order to filter out high-frequency variability and to highlight seasonal variability.





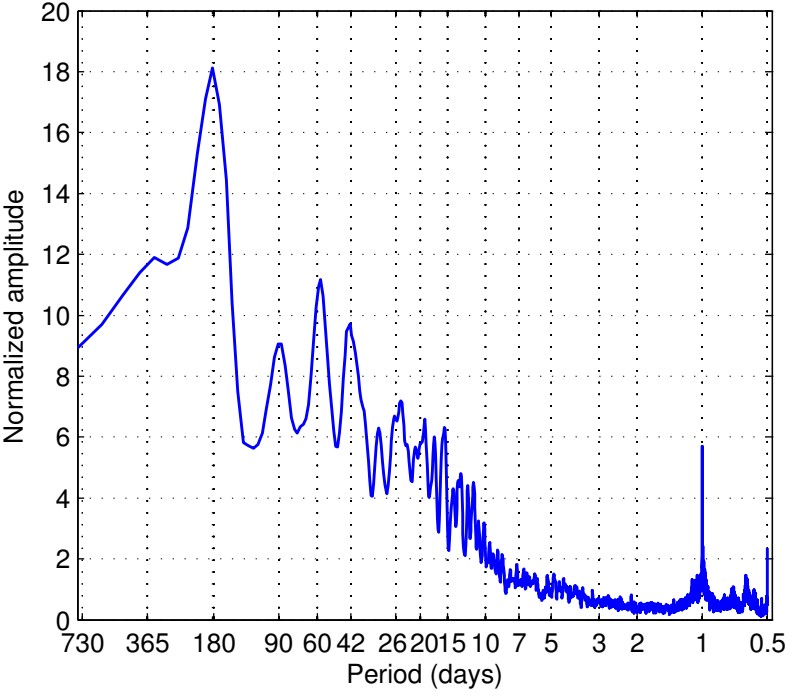

**Figure 6.** Frequency spectra of Kelvin wave energy for the zonal wavenumber $k = 1$ and summed over all vertical modes. A 1-2-1 filter with a Daniell kernel has been used to smooth the initial raw power spectra.

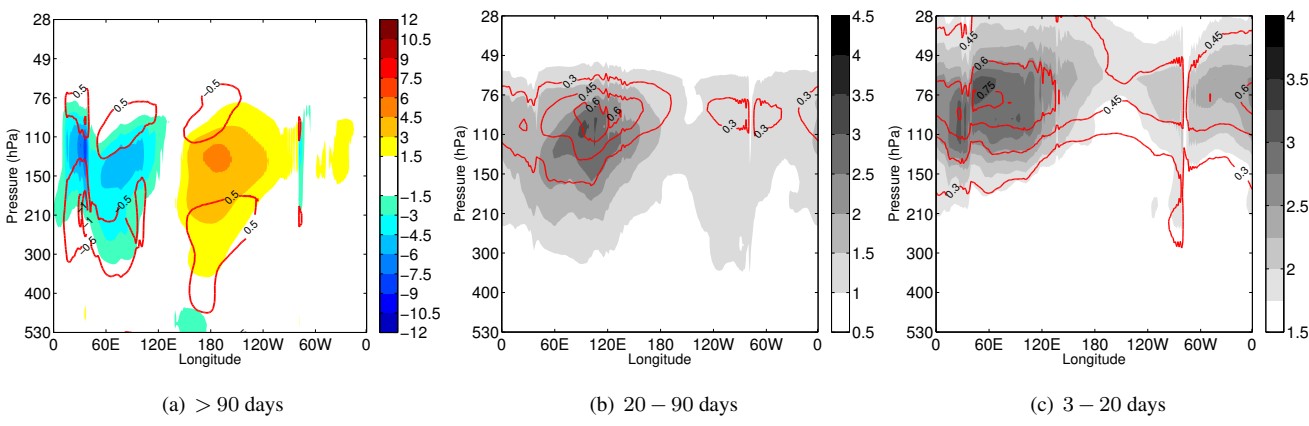

(a) $> 90$ days    (b) $20 - 90$ days    (c) $3 - 20$ days

**Figure 7.** Longitude-pressure sections along $0.7°$N of KW fields averaged over the full data period for: (a) low-frequency, (b) intraseasonal, and (c) intramonthly Kelvin wave variability with the associated filtered wave periods given in the panel captions. The contour scales are as follows: (a) mean KW zonal wind (colored fields, each $1.5$ ms$^{-1}$) and KW temperature (red contours, each $0.5$ K) components, (b) mean absolute amplitudes of KW zonal wind (grey fields, each $0.5$ ms$^{-1}$) and temperature (red contours, each $0.15$ K), and (c) mean absolute amplitudes of zonal wind (grey fields, each $0.25$ ms$^{-1}$) and temperature (red contours, each $0.15$ K).





(a) DJF

(b) MAM

(c) JJA

(d) SON

**Figure 8.** Longitude-pressure sections averaged over the following NH seasons: northern (a) winter, (b) spring, (c) summer, and (d) autumn. Kelvin wave zonal wind (blue-to-red colored contours) and temperature (red contours) fields on equator $0.7°$N have same contour values as in Fig. 7(a). The background zonal wind is shown in blue contours (each $5$ ms$^{-1}$, starting from $15$ ms$^{-1}$). A single static stability contour of value $5 \times 10^{-4}$ s$^{-2}$ is shown as a thick dotted black line in order to represent the seasonal movement of the tropical tropopause height. Both background wind and stability fields are latitudinally-averaged over $5°$S-$5°$N. All fields are smoothed a priori using a low-pass filter with cut-off period of 90 days.





**Figure 9.** Longitude-time section at model level 45 (∼ 153 hPa) of Kelvin wave zonal wind (blue to red shaded contours, each $2\,\mathrm{ms^{-1}}$) using values at $0.7°$N as well as Outgoing Longwave Radiation (red contours, each $10\,\mathrm{Wm^{-2}}$ starting at $225\,\mathrm{Wm^{-2}}$) averaged over latitudes from $15°$S till $15°$N. Both fields have been filtered a priori using a low-pass filter with cut-off at 90 day period.





(a) DJF

(b) JJA

(c) DJF - $N^2$ anomalies

(d) JJA - $N^2$ anomalies

**Figure 10.** Seasonal-averaged view for (a and c) DJF and (b and d) JJA months as function of longitude and pressure for the following fields: (a-b panels) KW temperature, $\overline{T_{kw}}^s$, (blue-to-red, each 0.25 K) and static stability field, $\overline{N^2}^s$ (black contours, each $1 \times 10^{-4}$ s$^{-2}$, starting at $2 \times 10^{-4}$ s$^{-2}$), (c-d panels) KW static stability anomaly, $\overline{N_{kw}^2}^s$ (blue-to-red, each $0.2 \times 10^{-4}$ s$^{-2}$), and static stability anomaly with respect to the zonal mean, $\overline{N'^2}^s$ (red contours, each $0.4 \times 10^{-4}$ s$^{-2}$).







**Figure 11.** Longitude-pressure sections along the equator ($0.7°$N) of intraseasonal Kelvin wave zonal wind (white-to-black shades, each $0.5$ ms$^{-1}$) and temperature (red contours, each $0.2$ K) fields averaged over the following seasons: NH (a) winter, (b) spring, (c) summer, and (d) autumn. Seasonal-averaging is performed over the absolute values of KW zonal wind and temperature. The background zonal wind (blue contours) and the tropical tropopause height (single thick dotted contour) are defined as in Fig. 8.



(a) DJF

(b) MAM

(c) JJA

(d) SON

**Figure 12.** Longitude-pressure sections along the equator ($0.7°$N) of intramonthly Kelvin wave zonal wind (white-to-black shades, each $0.25$ ms$^{-1}$) and temperature (red contours, each $0.2$ K) fields averaged over the following seasons: NH (a) winter, (b) spring, (c) summer, and (d) autumn. Seasonal-averaging is performed over the absolute values of KW zonal wind and temperature. The background zonal wind (blue contours) and the tropical tropopause height (single thick dotted contour) are defined as in Fig. 8.



(a) $|u'_{kw}| - 110$ hPa

(b) $|T'_{kw}| - 110$ hPa

(c) $|u'_{kw}| - 200$ hPa

(d) $|T'_{kw}| - 200$ hPa

**Figure 13.** Climatological view of subseasonal KW components (a and c) zonal wind, $|u'_{kw}|$ (blue-to-red shading, each $0.5$ ms$^{-1}$) and (b and d) temperature $|T'_{kw}|$ (blue-to-red shading, each $0.1$ K) as function of month in a calendar year and longitude along the equator ($0.7°$N) at (a and b) model level 40 ($\sim 110$ hPa) and (c and d) model level 49 ($\sim 200$ hPa). This includes all traveling KWs with periods $3 - 20$ days. For comparison, the background zonal wind field is illustrated as well (red contours, each $5$ ms$^{-1}$) at corresponding model levels. On the right side, the circle-marked blue line represents monthly mean values (over all $6+$ years) of daily maximum wave amplitudes occurring in longitude on the equator.