# Peer review of "Multivariate analysis of Kelvin wave seasonal variability in ECMWF L91 analyses"

_Atmospheric Chemistry and Physics, 2018_

## Referee Comment (RC1) · G. Kiladis (Referee) · 16 Mar 2018

Review of "Multivariate analysis of Kelvin wave seasonal variability in ECMWF L91 analyses", by Marten Blaauw and Nedjeljka Zagar, submitted to Atmos. Chem. Phys.

Minor revisions

General comments

This is a fine paper on Kelvin activity that utilizes the normal mode function decomposition method pioneered by Kasahara and Puri and further refined by Zagar and others. The paper essentially represents a "proof of concept" of the technique as applied to Kelvin waves, although a lot of interesting information is included. The paper succeeds in demonstrating the utility of NMF decomposition and should provide a good start-

ing point for those interesting in pursuing this approach, especially given the fact that software has been conveniently set up for others to apply as described in Zagar et al. 2015.

Specific comments

The paper appears to be in good shape overall, although there are some lingering questions in this reviewer's mind about interpretation, as discussed below. This mainly has to do with lumping all of the vertical modes together, which does not necessarily seem physical to me for all the cases considered. Perhaps not for this study, but it would be very instructive and add a lot of value to come up with some associated relationships between the subseasonal variability in Kelvin energy discussed here and some indices of tropical convection. The authors have made an initial attempt of this for the seasonal cycle and interannual timescale and their interpretation seems reasonable there. As far as higher frequencies go, for instance, cross spectra between the timeseries shown in Fig. 5 and geographically distributed OLR or brightness temperature could be very revealing. Ultimately, this could also be done for other modes isolated by this technique too. Figure captions could be improved overall, especially at the locations noted below. Also the text is not as clear as it could be in places.

Technical corrections

Comments by line number:

32: do you mean "modulate the TTL"?

49: nomenclature here is slightly confusing: hkw has not been defined. If you are indeed following Holton then this is a perturbation term, otherwise it might be mistaken for the equivalent depth.

58: this statement is misleading, actually the tropical "cloud activity" really refers to the mean cloudiness (Tindall et al. were citing Zhang 1993) which is a maximum in January and minimum in July near the equator.

79: the horizontal and temporal resolution from line 106 should also be included here.

93: not sure what "denotedm" means here, should this be in parentheses (denoted m)?

126: I am confused by one aspect of this procedure: As nicely discussed in detail by Zagar et al. (2015) and references therein, each vertical mode is characterized by an equivalent depth and associated horizontal structure function. Here it appears that all of the vertical modes are summed. For the sake of discussion, suppose we have a stratospheric "free" Kelvin mode which is present at the same time as an independent "convectively coupled" Kelvin mode that has maximum amplitude in the troposphere. These could be either collocated in lat-lon space or present at the same time in different regions of the globe. I think it would be profitable to make clear that it would still be possible to separate these modes by this procedure if one had enough additional information on the associated equivalent depths of each mode, which could be much different from each other. Stratospheric Kelvin waves at 50hPa follow dispersion related to a 120 m equivalent, whereas this is more like 25 m for convectively coupled waves. Acknowledging this fact seems appropriate, along with perhaps some words on how it could be dealt with in practice. I wonder, for instance, if investigating time series of KW energy for individual vertical modes could be done in a systematic way, using an extension of the approach in Zagar and Franzke (2015)? I think it would add considerable value to add a short discussion on these points.

137: As in the previous comment, you are including the projection onto the vertical mode that corresponds to, say, the 10 km equivalent depth, which I would assume be more representative of an external Kelvin mode. Perhaps one way to look at this is to assume that there would be a "spectrum" of vertical modes for each situation depending on how much the data projects onto each individual mode. I think it would it be worth elaborating on this point here, especially for those who may be less familiar with the idea that you are discretizing the vertical and associated horizontal structures for a reason, but that in reality a given atmospheric disturbance will be composed of a potentially different combination of these from case to case.

138: probably should describe the figure in more detail first, such as the tilted structures of what fields are shown, etc., before launching into the implications.

145: "strong KW activity was present" I guess you are referring to the case discussed in Fig. 2? Are you saying that the entire pattern shown represents a free Kelvin mode? This is where some information on the associated convective activity might be very useful.

154: Another very useful bit of information would relate the activity of KWs (such as measured by the energy spectrum) to the QBO, perhaps in a future study.

161: "climatological zonal structure" at first, I thought this was only for the period of Fig. 2 and the figure caption does not help.

191: "resonates" => "fluctuates" might be a better choice of words.

244: I'm unsure what the tidal effect would look like, but could the tide itself be projecting onto the Kelvin structure in some way? It doesn't seem that a Kelvin wave structure should be impacted by the tide, especially if you consider that these are both orthogonal "normal modes". This may deserve a few more words and could emphasize one potential drawback of the approach.

259: Probably should use a bit more care when discussing the impact of the QBO since this is highly dependent on the level you are referring to. One way to put it might be to point out that vertical penetration of Kelvin wave energy into the stratosphere depends on the state of the QBO in the lower layers of the stratosphere.

269: It turns out the the QBO was easterly at 50 hPa in July 2007, but only just beginning the easterly phase at that level.

291: What is the advantage of using "absolute amplitude" over say, variances? This should be discussed and justified.

287: I wouldn't insist on it, but it may also be of interest to examine whether there is

significant skewness in the distributions of the raw filtered data on the subseasonal timescales, and whether they are approximately normal. In other words, does it matter what the phase of the Kelvin component is at a given level, or are negative perturbations approximately the opposite of positive ones?

292: caption and discussion of Fig. 7 is confusing. When describing 7a the caption says "mean" when you really mean "(semi)annual" as discussed above.

Isn't 7c for the high frequency? If so the caption should say so.

297: "quadrupole" is more commonly used, but perhaps not in Europe.

310: There is decent agreement between the results here and those of Flannaghan and Fueglistaler as to where the Kelvin activity is maximized and I think this should be discussed at this point as well as below (e.g. Fig. 6 of Flannaghan and Fueglistaler 2013).

333: This is a very interesting analysis and the link between the low frequency Kelvin component to the "Gill-type response" is very insightful. However, the pure Gill response is only Kelvin-like to the east and includes Rossby gyres to the west of the heating, so at least part of the easterlies would not necessarily be due to a projection on Kelvin waves. This should at least be mentioned, if not discussed in more detail.

356: This is a clever way of getting at the impact of the quasi-stationary Kelvin wave forcing.

384: It would indeed be interesting to see what the relationship between intraseasonal Kelvin activity isolated here might be to the convective activity on the same timescale.

416: It seems that the right panels refer to a specific longitude only but this is never identified in the text or caption.

431: Now it seems that these numbers are somehow zonal averages? Rather confusing, and once again the figure caption does not help either.

437: I guess this is not shown? That should be noted.

Signed,

George Kiladis

––––––––––––––––––––––––––––––––

---

## Referee Comment (RC2) · Anonymous Referee #2 · 26 Mar 2018

\*\*Recommendation: Major revisions

\*\*General comments:

The authors have developed a powerful analysis technique whereby they are able to decompose any 3-dimensional atmospheric analysis product into its (linear) global normal modes, which includes the equatorial Kelvin wave as one of its components. As I understand it, this decomposition is computed for each individual time point of the analysis, and no information on the propagation from one time point to the next is used for the categorization into the different normal modes. This is quite different to what has been done in many other studies, for example, Wheeler and Kiladis (1999) who used wavenumber-frequency spectra and filtering for identification of equatorial waves. Therefore, what is called a "Kelvin wave" in this study is somewhat different to those

other studies, since the identified structures may not be propagating, but may be stationary or even display propagation in the opposite direction to what is usually ascribed to a particular mode. This difference with other studies requires careful explanation and should be highlighted, but is not necessarily a problem with the paper.

Another aspect of this work that I think needs highlighting is that the normal mode decomposition is based on the assumption that the equations of motion are linearized about a basic state of rest (i.e. zero winds, line 88). It is unclear to me how much this assumption may affect the results.

I also wonder what assumptions are made about the static stability for the calculation of the normal modes. The static stability is important for setting the relationship between the horizontal and vertical structures of the normal modes. For the same gravity wave speed, c, a Kelvin wave in the stratosphere will have a shorter vertical wavelength than a Kelvin wave in the troposphere, due to the different static stability. But both these Kelvin waves will have the same meridional (horizontal) structure, which is set by the equatorial Rossby radius, a function of g. So what temperature and static stability profiles do you assume, and how can this affect the results? What would happen if you assumed a "moist static stability" for the troposphere? Instead of the traditional dry static stability?

To be more convinced about the utility of the technique for understanding, I also wonder what the wavenumber-frequency spectra of the decomposed "Kelvin waves" would look like. You could do this at each level and see what equivalent depth dominates at each level. I imagine that in the troposphere you may see a predominance of the MJO and the c=∼20m/s convectively-coupled Kelvin waves, but as you enter the stratosphere the equivalent depth should start increasing due to the filtering provided by the background winds. These results would be useful to compare to Hendon and Wheeler (2008, J. Atmos. Sci, Vol 65).

Perhaps another interesting comparison to make is how the transient behaviour in OLR

matches the transient behaviour in your Kelvin wave dataset. Do the convectively-coupled Kelvin waves identified in OLR by the technique of Wheeler and Kiladis (1999) show up in your independent Kelvin wave dataset? To me, this would be much more interesting than some of the analysis provided here.

In summary, I must admit that I was a little underwhelmed by the results presented here. I think more interesting things could have been studied. But at the same time, the work is rigorous and may be more interesting to others, so it still adds something to the published literature.

**Specific comments:

Line 5. Why do you call it a "barotropic" KW response? Shouldn't this be the baroclinic mode with a half-sinusoid vertical structure in the troposphere?

Line 64. Missing "the" before "information".

Lines 74 or 75. Change to "covers approximately 6.5 years from January 2007 until June 2013".

Line 93. "denotedm"?

Lines 104-105. It is confusing to me to denote the KW as the n=0 EIG mode, since in many other papers (e.g. Matsuno 1966 and Wheeler and Kiladis 1999) the n=0 mode is the continuation of the mixed Rossby-gravity mode through the wavenumber 0 axis. In these papers the KW is the n=-1 solution.

Lines 138-139. I found this difficult to read because of the use of parentheses to provide the opposite meaning – please read the paper https://eos.org/opinions/parentheses-are-are-not-for-references-and-clarification-saving-space

Line 141. "zonal wind" not "zonal wave".

Line 146 and many other locations. Add "the" before "Eastern hemisphere".

[Figure]

Figure 4. I didn't find this figure to be very informative. A wavenumber-frequency spectrum of the Kelvin wave dataset at a few different vertical levels would have been more interesting.

Line 221 and many other locations. What is "summer" at the equator? It doesn't make sense to call the seasons using "summer", "autumn", "winter", and "spring" for equatorial waves. I would prefer you just call them "DJF", "MAM", "JJA", "SON".

Line 260. You say "when the ENSO index is positive". Do you mean "during El Nino"?

Lines 262-265. It is perhaps also important to note that the MJO was quite strong in 2007-08 (e.g. as defined by the Real-time Multivariate MJO index), and that the MJO has been found to be generally stronger in easterly QBO years (Sun et al. 2017). I am also fairly certain that the MJO must project quite stronger onto your Kelvin wave mode.

Line 298. I think you mean "warm anomalies", not "heating".

Line 305. Why do you call these intramonthly KWs the "free propagating" waves? If "free" means away from the forcing of convection, then isn't every wave in the strato-sphere "free"?

Line 385. Please call this section "Intramonthly propagating Kelvin waves".

Figure 13. I found this very difficult to understand. On line 415 you say "different years", but what different years"? Is this a composite of all years? On line 416 you say "specific longitude". What specific longitude? The caption says it is a "climatology", but why is it so noisy if it is a climatology?

Figure 1 caption. Remove text "(panel b in Fig. 1)"

Figure 5. There appears to be some data missing at the end of 2009.

---

## Referee Comment (RC3) · Anonymous Referee #2 · 27 Mar 2018

On page C2 of my review it says "a function of g". I meant "a function of c", where c is the gravity wave speed. The meridional length scale is actually sqrt(c/Beta), where Beta is df/dy.

I am sorry for this error.

———————————————

---

## Author Comment (AC1) · 8 Apr 2018

April 2018

Response to Dr George Kiladis (Reviewer 1) comments on

"Multivariate analysis of Kelvin wave seasonal variability in ECMWF L91 analyses"

by Marten Blaauw and Nedjeljka Žagar

Dear Dr Kiladis,

Thank you very much for your detailed and constructive comments on the paper.

We have taken them into account in the revised paper.

We hope that the revised paper better highlights analysis possibilities offered by the MODES package i.e. by the multivariate analysis of the Kelvin wave and other equatorial waves on the sphere.

Enclosed please find our responses to your comments using the same organisation as in the review.  Your comments are coloured blue whereas our responses are in black.

Yours sincerely,

Marten Blaauw and Nedjeljka Žagar

**GENERAL COMMENT**:

*This is a fine paper on Kelvin activity that utilizes the normal mode function decomposition method pioneered by Kasahara and Puri and further refined by Zagar and others. The paper essentially represents a "proof of concept" of the technique as applied to Kelvin waves, although a lot of interesting information is included. The paper succeeds in demonstrating the utility of NMF decomposition and should provide a good starting point for those interesting in pursuing this approach, especially given the fact that software has been conveniently set up for others to apply as described in Zagar et al. 2015.*

**SPECIFIC COMMENTS:**

*The paper appears to be in good shape overall, although there are some lingering questions in this reviewer's mind about interpretation, as discussed below. This mainly has to do with lumping all of the vertical modes together, which does not necessarily seem physical to me for all the cases considered. Perhaps not for this study, but it would be very instructive and add a lot of value to come up with some associated relationships between the subseasonal variability in Kelvin energy discussed here and some indices of tropical convection. The authors have made an initial attempt of this for the seasonal cycle and interannual timescale and their interpretation seems reasonable there. As far as higher frequencies go, for instance, cross spectra between the timeseries shown in Fig. 5 and geographically distributed OLR or brightness temperature could be very revealing. Ultimately, this could also be done for other modes isolated by this technique too. Figure captions could be improved overall, especially at the locations noted below. Also the text is not as clear as it could be in places.*

**Response**: Thank you very much for the comments and suggestions. Indeed, the first goal of the paper is to introduce a novel methodology to analyze linear Kelvin waves on the sphere. As shown by Boyd and Zhou (J. Atmos. Sci., 2008), the degree of the Kelvin wave equatorial confinement on the sphere is controlled not only by the equivalent depth (i.e. by the equatorial Rossby radius of deformation) as on the equatorial beta plane but also by the zonal wavenumber. Therefore, even barotropic Kelvin waves with equivalent depth around 10 km are on the sphere trapped to the equator. This is a strong reason to use the spherical Kelvin wave solution for the projection.

We demonstrate the method by examples and diagnostics of seasonal variability that leads to results that are in agreement with previous studies, but also provide new understanding. As the first paper using this method for the Kelvin wave analysis, it takes space to present how the modal decomposition works and what kind of outputs it provides. As you pointed out, there are numerous research questions where this method may add insight and complement existing diagnostics. We find it hard to expand the paper beyond what has been already included. One of the topics left for future work is the suggested cross spectra.

**Technical corrections**

*32: do you mean "modulate the TTL"?*

**Response:** Yes. Corrected.

*49: nomenclature here is slightly confusing: hkw has not been defined. If you are indeed following Holton then this is a perturbation term, otherwise it might be mistaken for the equivalent depth.*

**Response:** The paragraph has been reformulated. We clarify that h_kw is the geopotential height perturbation associated with the Kelvin wave.

*58: this statement is misleading, actually the tropical "cloud activity" really refers to the mean cloudiness (Tindall et al. were citing Zhang 1993) which is a maximum in January and minimum in July near the equator.*

**Response:** The sentence has been changed as follows:

Such analysis was carried out by \citet{Tindall2006b} for the lower stratosphere for the ERA-15 data in 1981-93 period. Their results suggested that KWs contributes approximately 1 K$^2$ of the temperature variance on the equator with peak activity occurring during solstice seasons at 100 hPa, during December$-$February at 70 hPa and at 50 hPa it occurs during the easterly to westerly quasi-biennial oscillation (QBO) phase transition.

*79: the horizontal and temporal resolution from line 106 should also be included here.*

**Response:** Changed as suggested.

*93: not sure what "denotedm" means here, should this be in parentheses (denoted m)?*

**Response:** We are sorry for the typo. It has been corrected.

*126: I am confused by one aspect of this procedure: As nicely discussed in detail by Zagar et al. (2015) and references therein, each vertical mode is characterized by an equivalent depth and associated horizontal structure function. Here it appears that all of the vertical modes are summed. For the sake of discussion, suppose we have a stratospheric "free" Kelvin mode which is present at the same time as an independent "convectively coupled" Kelvin mode that has maximum amplitude in the troposphere. These could be either collocated in lat-lon space or present at the same time in different regions of the globe. I think it would be profitable to make clear that it would still be possible to separate these modes by this procedure if one had enough additional information on the associated equivalent depths of each mode, which could be much different from each other. Stratospheric Kelvin waves at 50hPa follow dispersion related to a 120 m equivalent, whereas this is more like 25 m for convectively coupled waves. Acknowledging this fact seems appropriate, along with perhaps some words on how it could be dealt with in practice. I wonder, for instance, if investigating time series of KW energy for individual vertical modes could be done in a systematic way, using an extension of the approach in Zagar and Franzke (2015)? I think it would add considerable value to add a short discussion on these points.*

**Response:** We deliberately use all vertical modes summed up their sum provides the total Kelvin wave signal in physical space. We do use a smaller number of vertical modes (M=60) then the number of levels with data (91) but the decomposition provides the complete information about the waves in the studied layer of upper troposphere and above. Because the majority of the results concerns the total KW signal in physical space we do not make specific references to equivalent depths in the discussion.

In the revised paper, we add some more discussion about the role of equivalent depth in the construction process of the projection. It is not straightforward to discuss results on any level in terms of equivalent depths as for this we would need information on the amount of the Kelvin wave signal projecting to various equivalent depths. This can be obtained by filtering to physical space each vertical mode separately that increases the computational demand by factor $m$. In this study we limited the discussion to the basic concept of the method and seasonal features of the complete Kelvin wave signal.

It would be possible to split the Kelvin wave filtering in terms of vertical modes and discuss how (if) the KW signal in terms of m is grouped in various ranges of $m$ (as indicated for MJO-related variance in Žagar and Franzke (GRL, 2015). Such diagnosis is less trivial in the case with the high model lid such as here (1 Pa). The top model levels in this period also had some artificial damping and wave reflections. One way to solve this would be to limit the NMF projection to the troposphere only or the troposphere+lower stratosphere as for example done by Žagar et al (2017, JAS) using the ERA Interim data. The ongoing work on the KW properties in reanalysis datasets should provide further insight which we hope to report soon.

*137: As in the previous comment, you are including the projection onto the vertical mode that corresponds to, say, the 10 km equivalent depth, which I would assume be more representative of an external Kelvin mode. Perhaps one way to look at this is to assume that there would be a "spectrum" of vertical modes for each situation depending on how much the data projects onto each individual mode. I think it would it be worth elaborating on this point here, especially for those who may be less familiar with the idea that you are discretizing the vertical and associated horizontal structures for a reason, but that in reality a given atmospheric disturbance will be composed of a potentially different combination of these from case to case.*

**Response:** The revised paper includes some more discussion of the role of vertical decomposition i.e. the equivalent depth. We do not filter any vertical mode separately as it would broaden presentation beyond the scope of this paper. Earlier paper by Žagar et al (Mon. Wea. Rev., 2009b) showed that presentation in terms of vertical modes in time can be very useful to represent the vertical Kelvin wave propagation. It is another topic left for future studies.

Our preliminary results on how the KW energy projects indeed among a "spectrum" of vertical modes, divides the vertical modes into roughly two groups: the vertical modes with equivalent depths ranging from 10 to 0.1 km that represent a part of the signal characterized by a strong (semi)annual periodicity. The second group contains signal that project to waves with equivalent depths ranging from 100 to 10 m and are observed throughout the year. The physical signature of these waves, decomposed based on their numerically discretized vertical structure functions, is not well understood yet in relation to the free stratospheric and the convectively coupled KWs that you mentioned in the comment. As for the previous comment, a dataset with a lower lid such as reanalysis data would make the interpretation task easier.

138: probably should describe the figure in more detail first, such as the tilted structures of what fields are shown, etc., before launching into the implications.

**Response:** We changed section 2.2 to provide more clarity on the Kelvin wave examples. In Fig. 2 different July days in 2010 have been chosen (25, 28, 31 July) with a shorter time interval of three days in order to visualize the stratospheric KW propagation more clearly. The chronological order of explanation has changed in section 2.2 such that Figures 1 and 2 are described in more detail first with respect to the free stratospheric waves as well as quasi-stationary tropopause structures, followed by the implications.

145: "strong KW activity was present" I guess you are referring to the case discussed in Fig. 2? Are you saying that the entire pattern shown represents a free Kelvin mode? This is where some information on the associated convective activity might be very useful.

**Response:** Yes. In Fig. 2 we refer in particular to 28 and 31 July when the wave package is located in the Eastern Hemisphere. It is hard to argue that the whole stratospheric pattern represents a free Kelvin mode. We rewritten the discussion. There are clear signs of eastward and downward KW propagation above 80 hPa. It is hard to see visually where coupled Kelvin modes in the TTL end and where the free Kelvin modes start to dominate.

To clarify the relationship between the observed Kelvin wave activity in relation to the background wind and stratification conditions, we illustrate below the background fields as well:

[Figure]

(a) 25-07-2010                     (b) 28-07-2010                     (c) 31-07-2010

The bottom three panels are as in revised Fig. 2 in the paper. The top three panels illustrate from the ECMWF zonal wind (blue-to-red shades, for |U| > 10 ms$^{-1}$) and the static stability (red contours, for $N^2 > 2 \times 10^{-4} s^{-2}$) fields. The easterly winds in the TTL (160 hPa) create a "window" through which the Kelvin wave energy can "escape" into the stratosphere. A double-folding structure in the static stability fields (up to $8 \times 10^{-4} s^{-2}$) coincides with large amplitude Kelvin wave temperature perturbations (up to 4 K).

154: Another very useful bit of information would relate the activity of KWs (such as measured by the energy spectrum) to the QBO, perhaps in a future study.

**Response:** We agree and the Kelvin wave-QBO relation is one of the couplings we hope to address using a longer time series of data from reanalyses.

161: "climatological zonal structure" at first, I thought this was only for the period of Fig. 2 and the figure caption does not help.

**Response:** The caption has been re-written. Figure 3 shows the six-year average of full zonal wind in the analyses and static stability.

191: "resonates" => "fluctuates" might be a better choice of words.

**Response:** Changed as suggested.

244: I'm unsure what the tidal effect would look like, but could the tide itself be projecting onto the Kelvin structure in some way? It doesn't seem that a Kelvin wave structure should be impacted by the tide, especially if you consider that these are both orthogonal "normal modes". This may deserve a few more words and could emphasize one potential drawback of the approach.

**Response:** Rewritten as "The spectrum contains a peak at 1-day period associated with the diurnal tide partially projecting on the Kelvin waves." without further elaboration which is beyond the scope of present study. An unpublished master thesis used MODES to analyze what large-scale wave the tides in the same dataset project on.

259: Probably should use a bit more care when discussing the impact of the QBO since this is highly dependent on the level you are referring to. One way to put it might be to point out that vertical penetration of Kelvin wave energy into the stratosphere depends on the state of the QBO in the lower layers of the stratosphere.

**Response:** Corrected as: "Moreover, KW activity is enhanced whenever easterly QBO winds are present down into the lower stratosphere \citep{Baldwin2001, Alexander2010} or during El Ni$\tilde{\text{n}}$o \citep{Yang2013}."

269: It turns out the the QBO was easterly at 50 hPa in July 2007, but only just beginning the easterly phase at that level.

**Response:** Thank you. We have added this comment.

291: What is the advantage of using "absolute amplitude" over say, variances? This should be discussed and justified.

**Response:** The explanation has been added as the following footnote in the revised manuscript: "Most previous studies define KW activity as square amplitude rather than absolute amplitude. In our high resolution dataset we observe highly localized patterns of the KW activity in the Eastern hemisphere due to ongoing wave amplification. By using absolute amplitudes we better visualize the longitudinal structure of the KW activity in comparison to its local maxima."

287: I wouldn't insist on it, but it may also be of interest to examine whether there is significant skewness in the distributions of the raw filtered data on the subseasonal timescales, and whether they are approximately normal. In other words, does it matter what the phase of the Kelvin component is at a given level, or are negative perturbations approximately the opposite of positive ones?

**Response:** To answer your question, we checked the distribution of raw filtered data with applied low-pass filter over periods of 3-20 days for the KW zonal wind and temperature components at several levels. Enclosed are examples for (left panel) KW zonal wind at ~153 hPa (in ms$^{-1}$) and (right panel) KW temperature at ~71 hPa (in K). All scatter point values over 6-year period are plotted as a function of longitude. The 6-year mean (red line) and median (green line) are computed for each longitude.

It follows that the distribution is well-described as normal on subseasonal timescales. Mean and median values are approximately zero.

The same applies for filtered data for periods 20-90 days and for other levels.

[Figure]

292: caption and discussion of Fig. 7 is confusing. When describing 7a the caption says "mean" when you really mean "(semi)annual" as discussed above. Isn't 7c for the high frequency? If so the caption should say so.

**Response:** Figure caption has been rewritten. The word "mean" has been removed. All three panels represented 6-year mean Kelvin wave zonal wind and temperature fields that are filtered over three different specific ranges of periods.

297: "quadrupole" is more commonly used, but perhaps not in Europe.

**Response:** Changed as suggested.

310: There is decent agreement between the results here and those of Flannaghan and Fueglistaler as to where the Kelvin activity is maximized and I think this should be discussed at this point as well as below (e.g. Fig. 6 of Flannaghan and Fueglistaler 2013).

**Response:** This is discussed in details in section 4.4 where seasonal decomposition similar to Figure 6 of Flannaghan and Fueglistaler (2013) is presented.

333: This is a very interesting analysis and the link between the low frequency Kelvin component to the "Gill-type response" is very insightful. However, the pure Gill response is only Kelvin-like to the east and includes Rossby gyres to the west of the heating, so at least part of the easterlies would not necessarily be due to a projection on Kelvin waves. This should at least be mentioned, if not discussed in more detail.

**Response:** The Gill-type response results from the steady-state, long-wave approximation without background wind that leads to an idealized view with Kelvin waves represented only by westerlies east of the Maritime continent. Real data is more complex and we have been more careful in the revised paper in using the "Gill-type KW" term following Salby and Garcia (1987). We emphasize that we discuss the low-frequency Kelvin wave response to large-scale tropical heating that has strong easterly zonal wind component as well.

356: This is a clever way of getting at the impact of the quasi-stationary Kelvin wave forcing.

**Response:** Thank you

384: It would indeed be interesting to see what the relationship between intraseasonal Kelvin activity isolated here might be to the convective activity on the same timescale.

**Response:** We recognize this as an interesting question but beyond the scope of the present paper.

416: It seems that the right panels refer to a specific longitude only but this is never identified in the text or caption.

**Response:** Figure caption has been re-written and discussion improved. The amplitude of the KW zonal wind wave are maximal amplitudes occurring anywhere along the equator averaged over the 6-year period for each calendar month.

431: Now it seems that these numbers are somehow zonal averages? Rather confusing, and once again the figure caption does not help either.

**Response:** Corrected. These numbers are maximal values found along the equator.

437: I guess this is not shown? That should be noted.

**Response**:  Added as suggested.

[revised manuscript text omitted]

---

## Author Comment (AC2) · 8 Apr 2018

Dear Referee,

thank you. We have included this correction in your main comments.

Sincerely,

Nedjeljka Zagar

---

## Author Comment (AC3) · 8 Apr 2018

April 2018

Response to the comments of Reviewer 2 on

"Multivariate analysis of Kelvin wave seasonal variability in ECMWF L91 analyses"

by Marten Blaauw and Nedjeljka Žagar

Dear Referee, thank you very much for your comments and suggestion on our manuscript.

We have revised paper following your criticism and suggestions. In particular, we have re-written parts of Introduction and methodology sections in order to better describe novel features of the applied method.

Enclosed please find our responses to your comments using the same organisation as in your review.  Your comments are coloured blue whereas our responses are in black.

Your sincerely,

Marten Blaauw and Nedjeljka Žagar

**GENERAL COMMENTS**:

*The authors have developed a powerful analysis technique whereby they are able to decompose any 3-dimensional atmospheric analysis product into its (linear) global normal modes, which includes the equatorial Kelvin wave as one of its components. As I understand it, this decomposition is computed for each individual time point of the analysis, and no information on the propagation from one time point to the next is used for the categorization into the different normal modes. This is quite different to what has been done in many other studies, for example, Wheeler and Kiladis (1999) who used wavenumber-frequency spectra and filtering for identification of equatorial waves. Therefore, what is called a "Kelvin wave" in this study is somewhat different to those C1 ACPD Interactive comment Printer-friendly version Discussion paper other studies, since the identified structures may not be propagating, but may be stationary or even display propagation in the opposite direction to what is usually ascribed to a particular mode. This difference with other studies requires careful explanation and should be highlighted, but is not necessarily a problem with the paper.*

*Another aspect of this work that I think needs highlighting is that the normal mode decomposition is based on the assumption that the equations of motion are linearized about a basic state of rest (i.e. zero winds, line 88). It is unclear to me how much this assumption may affect the results.*

*I also wonder what assumptions are made about the static stability for the calculation of the normal modes. The static stability is important for setting the relationship between the horizontal and vertical structures of the normal modes. For the same gravity wave speed, c, a Kelvin wave in the stratosphere will have a shorter vertical wavelength than a Kelvin wave in the troposphere, due to the different static stability. But both these Kelvin waves will have the same meridional (horizontal) structure, which is set by the equatorial Rossby radius, a function of c, where c is the gravity wave speed. The meridional length scale is actually sqrt(c/Beta), where Beta is df/dy. So what temperature and static stability profiles do you assume, and how can this affect the results? What would happen if you assumed a "moist static stability" for the troposphere? Instead of the traditional dry static stability?*

*To be more convinced about the utility of the technique for understanding, I also wonder what the wavenumber-frequency spectra of the decomposed "Kelvin waves" would look like. You could do this at each level and see what equivalent depth dominates at each level. I imagine that in the troposphere you may see a predominance of the MJO and the c=∼20m/s convectively-coupled Kelvin waves, but as you enter the stratosphere the equivalent depth should start increasing due to the filtering provided by the background winds. These results would be useful to compare to Hendon and Wheeler (2008, J. Atmos. Sci, Vol 65).*

*Perhaps another interesting comparison to make is how the transient behaviour in OLR matches the transient behaviour in your Kelvin wave dataset. Do the convectivelycoupled Kelvin waves identified in OLR by the technique of Wheeler and Kiladis (1999) show up in your independent Kelvin wave dataset? To me, this would be much more interesting than some of the analysis provided here.*

*In summary, I must admit that I was a little underwhelmed by the results presented here. I think more interesting things could have been studied. But at the same time, the work is rigorous and may be more interesting to others, so it still adds something to the published literature*

**Response**
**1. Methodology**
The first goal of our paper is to introduce a novel methodology for the Kelvin wave filtering and demonstrate it by examples and diagnostics with similarity to previous studies.

We use analytical relationships for the horizontal structure of the Kelvin wave wind and geopotential height perturbations on the sphere. This is different from many previous studies which relied on the Kelvin wave solutions on the equatorial beta plane. We sum up linear Kelvin wave solutions in many shallow-water equation systems (60 in our case). The extension of the Kelvin wave analysis by 2D approach (on individual horizontal levels or vertical planes) to the three-dimensional (3D) spherical coordinates in an important step for realistic filtering of Kelvin waves in global datasets. As shown by Boyd and Zhou (J. Atmos. Sci., 2008), the degree of the Kelvin wave equatorial confinement on the sphere is controlled not only by the equivalent depth (i.e. by the equatorial Rossby radius of deformation) but also by the zonal wavenumber. Therefore, even barotropic Kelvin waves with equivalent depth around 10 km on the sphere are trapped to the equator. This is a strong reason to use the spherical Kelvin wave solution for the projection.

Data projection on horizontal Kelvin wave structure, along with other equatorial waves, on individual time instants was performed before by Tindall et al (2006, QJRMS). In the revised paper we add also a reference to a similar approach by Yang et al (J. Atmos. Sci., 2003) for data on individual levels using the equatorial beta plane solutions and using the equatorial Rossby deformation radius as the fitting parameter. We also refer to Žagar et al (QJRMS, 2005, 2007) who used analytical solutions on the equatorial beta plane to analyze the distribution of the equatorial wave variance in the short-term forecast errors of the ECWMF model.

Our Kelvin wave signal at any time is a sum of many Kelvin waves with different phase speeds. As the applied normal-mode function projection provides a complete projection basis, we can quantify the amount of total variance associated with Kelvin waves in global data.

We have re-written parts of Introduction and other sections in order to better highlight methodology approach and what kind of outputs it provides. We believe that here are many research questions regarding the Kelvin and other equatorial waves where the presented method can provide added insight and complement other methods.

**2. Comparison with spectral space-time filtering (so-called Wheeler-Kiladis diagrams)**
Our filtering procedure is different from the widely used spectral space-time filtering pioneering by Hayashi (1972) that applies Kelvin wave dispersion relations. We analyze circulation data at selected processing times independently of other times. In other words, the dispersion relationship for the linear Kelvin waves on the sphere, used to derive the analytical expressions for the Kelvin wave wind and geopotential height, is not used explicitly in the data analysis. At every analyzed time step, in every grid point we sum up contributions from 60 Kelvin wave solutions for each zonal wavenumber. Nevertheless, the linear wave features readily persist in our outputs for the Kelvin waves as shown in the example and Hovmoeller diagrams in the Result section. This result alone is very interesting as it validates the linear wave theory approach that has been successfully employed in many studies, especially for the large-scale tropical circulation features.

The spectral space-time filtering does not consider the meridional wave structure. The normal-mode function projection, thanks to its 3D orthogonal structures, allows a full quantification of the Kelvin wave signal and its  spatial localization.

We agree that would be interesting to combine the two different methods on analyzing Kelvin waves. But such analysis is beyond the scope of the present paper. Previously mentioned paper by Yang et al (J. Atmos. Sci., 2003) and several their follow-on studies combined the space-time filtering and the projection on analytical equatorial wave solution on the equatorial beta plane.  Our 6.5 year long dataset is also shorter than time series used in space-time filtering. We aim to perform some comparison of the two methods within the ongoing analysis of Kelvin waves in several decades long time series from reanalysis data.

**3. Linearization about a basic state of rest**

This is not a drawback of the method as wave frequencies are used solely for the formulation of the projection basis and not for studying the wave propagation properties. Namely, the frequencies differ depending on whether the linearization is performed around the state of rest as in our case or the mean flow is taken into account. If the mean zonal flow is taken into account, the frequencies of wave solutions can become unstable, as well as continuous, except for a few of the lowest balanced modes (Kasahara, 1980). Fortunately, the meridional structures of the Hough functions are not significantly different if the linearization is performed around the non-zero mean zonal flow (see Corrigendum to Kasahara, 1980, J. Atmos. Sci.). It is therefore suitable to use the Hough functions constructed with reference to the basic state at rest as a basis for the projection.

We have discussed this issue in the paper by Žagar et al. (2015, Geo. Model Dev.) where the projection method has been described in details. In Žagar et al. (2017, J. Atmos. Sci.) we demonstrated that even inertia-gravity waves with smaller scales can be successfully represented by the method. Another paper using the same decomposition, currently also in ACP Discussion, demonstrates the same point: https://www.atmos-chem-phys-discuss.net/acp-2018-228/

The impact of latitudinal shear on the Kelvin waves was previously shown negligible by Boyd (1978, J. Atmos. Sci.).

**4. Assumptions about the static stability for the calculation of the normal modes**

Stability and temperature are globally averaged and their temporal changes are not significant for the structure of the basis functions for the projection. In any case, definition of stability is a part of the derivation procedure by Kasahara and Puri (1981, Mon. Wea. Rev.) that considers hydrostatic atmosphere described by primitive equations. Moisture enters in the computation of geopotential on the terrain-following levels where virtual temperature is used.

We did not spend extra space on such details as it has been discussed in previous papers where we discussed the normal-mode function projection methodology. For example, Žagar et al. (2015, Geo. Model Dev.) showed typical vertical profiles of globally averaged temperature and stability which are input to the vertical structure equation. Normal-mode functions are derived for the bounded atmosphere with lid located at the model top half-level (pressure=0, sigma = 0). The same condition is applied in NWP models and indeed in the model of ECMWF that produced the analyzed data.

With the global surface temperature specified, the equivalent depth of the first vertical mode (barotropic mode) is always about 10 km as the barotropic equivalent depth depends only on the surface temperature and the atmosphere depth (Cohn and Dee, QJRMS, 1989). Analysis of Staniforth et al. (1985) showed that the equivalent depths for subsequent internal modes are relatively insensitive to the value of the surface boundary condition, but they are sensitive to the top boundary conditions (stability and the depth of the top model layers). This means that the vertical model depth matters for the shape of the vertical structure functions. This is the reason why we analyzed only their sum i.e. the total Kelvin wave signal and not individual vertical modes.

**SPECIFIC COMMENTS:**

*Line 5. Why do you call it a "barotropic" KW response? Shouldn't this be the baroclinic mode with a half-sinusoid vertical structure in the troposphere?*
**Response:** Thank you for noticing this typo. The abstract has been rewritten.

*Line 64. Missing "the" before "information".*
**Response:** Corrected. Thank.

*Lines 74 or 75. Change to "covers approximately 6.5 years from January 2007 until June 2013".*
**Response:** Changed as suggested.

*Line 93. "denotedm"?*
**Response:** This typo has been corrected.

*Lines 104-105. It is confusing to me to denote the KW as the n=0 EIG mode, since in many other papers (e.g. Matsuno 1966 and Wheeler and Kiladis 1999) the n=0 mode is the continuation of the mixed Rossby-gravity mode through the wavenumber 0 axis. In these papers the KW is the n=-1 solution.*
**Response:** We follow derivation and classification of wave solutions of the linearized shallow-water equations on the sphere from Žagar et al. (2015, Geo. Mod. Dev.) and references therein. Enclosed is figure 1 from the paper which shows dispersion curves for four equivalent depths.

[Figure]

**Figure 1.** Frequencies of spherical normal modes for different equivalent depths. (a) $D = 10\,\text{km}$, (b) $D = 1\,\text{km}$, (c) $D = 100\,\text{m}$ and (d) $D = 10\,\text{m}$. Frequencies are normalized by $2\Omega$ factor and shown in a logarithmic scale. Frequencies of the easterly and westerly inertia-gravity modes (EIG and WIG, respectively) are shown for the meridional modes $n = 0, 1, 3, 6, 9, 14, 19, 24, 29, 34, 39, 49, 59$ and $69$. For the balanced modes (ROT), meridional modes are shown for $n = 0, 1, 3, 5, 7, 9, 14, 19, 24, 29, 34, 39, 49, 59$ and $69$. Frequencies of the Kelvin modes ($n = 0$ EIG) and MRG modes ($n = 0$ ROT) are shown by magenta-coloured symbols. Frequencies of ROT modes $n > 1$ are denoted by grey circles and interconnected by dashed black lines. The EIG and WIG mode frequencies are shown by blue and red symbols, respectively. Negative frequencies correspond to negative values of zonal wave numbers. Frequencies for $k = 0$ are zero for all ROT modes, for the MRG mode, for the Kelvin mode and for the $n = 0$ WIG mode. For $n > 0$ and $k = 0$, frequencies of the WIG modes have opposite signs and equal values as the EIG mode frequencies. For $k > 1$, frequencies of the WIG modes have larger absolute values than frequencies of the EIG modes for the same $n$.

*Lines 138–139. I found this difficult to read because of the use of parentheses to provide the opposite meaning – please read the paper https://eos.org/opinions/parenthesesare-are-not-for-references-and-clarification-saving-space*
**Response:** It has been changed throughout the text.

*Line 141. "zonal wind" not "zonal wave".*
**Response:** Changed as suggested.

*Line 146 and many other locations. Add "the" before "Eastern hemisphere".*
**Response:** Corrected.

*Figure 4. I didn't find this figure to be very informative. A wavenumber-frequency spectrum of the Kelvin wave dataset at a few different vertical levels would have been more interesting.*
**Response:** In the spectral i.e. modal space we can not provide the Kelvin wave energy spectrum on individual levels as we perform vertical decomposition. At each level, our KW wind and temperature perturbations include contributions from 60 spherical shallow water models meaning 60 phase speeds. While it is different from previous studies of Kelvin wave spectrum, our spectrum quantifies the vertically integrated Kelvin wave total (potential + kinetic) energy in global data. This integrated energy depends on the vertical model depth. This is a property of the global normal-mode decomposition which may be most different from widely used single shallow-water equation system on the equatorial beta plane.

*Line 221 and many other locations. What is "summer" at the equator? It doesn't make sense to call the seasons using "summer", "autumn", "winter", and "spring" for equatorial waves. I would prefer you just call them "DJF", "MAM", "JJA", "SON".*
**Response:** Changed everywhere as suggested.

*Line 260. You say "when the ENSO index is positive". Do you mean "during El Nino"?*
**Response:** Yes. Changed as suggested.

*Lines 262-265. It is perhaps also important to note that the MJO was quite strong in 2007-08 (e.g. as defined by the Real-time Multivariate MJO index), and that the MJO has been found to be generally stronger in easterly QBO years (Sun et al. 2017). I am also fairly certain that the MJO must project quite stronger onto your Kelvin wave mode.*
**Response:** Thank you. We have updated text to include your comment on the strong MJO in this period of the strong KW activity.

*Line 298. I think you mean "warm anomalies", not "heating"*
**Response:** Changed as suggested.

*Line 305. Why do you call these intramonthly KWs the "free propagating" waves? If "free" means away from the forcing of convection, then isn't every wave in the stratosphere "free"?*
**Response:** We agree with the referee that the term "free propagating" KWs refers to the part of the Kelvin wave signal away from convective forcing, whereas in our case we discuss intramonthly KWs which are possibly coupled to convection. The revised paper uses the term "intramonthly KWs" to describe waves with periods 3-20 days.

*Line 385. Please call this section "Intramonthly propagating Kelvin waves".*
**Response:** Changed as suggested.

*Figure 13. I found this very difficult to understand. On line 415 you say "different years", but what different years"? Is this a composite of all years? On line 416 you say "specific longitude". What specific longitude? The caption says it is a "climatology", but why is it so noisy if it is a climatology?*
**Response:** The word climatology was not appropriate here. This figure is a composite of intramonthly Kelvin waves in all years as recognized by the reviewer. We have rewritten the caption and figure description in the text.

*Figure 1 caption. Remove text "(panel b in Fig. 1)"*
**Response:** Corrected

*Figure 5. There appears to be some data missing at the end of 2009.*
**Response:** The limits of x-axes in the figure were corrected. Thank you for noticing it.

[revised manuscript text omitted]